# Inherited *ARPC5* mutations cause an actinopathy impairing cell motility and disrupting cytokine signaling

Cristiane J. Nunes-Santos[1], HyeSun Kuehn [1], Brigette Boast [1], SuJin Hwang[1], Douglas B. Kuhns [2], Jennifer Stoddard[1], Julie E. Niemela[1], Danielle L. Fink[2], Stefania Pittaluga [3], Mones Abu-Asab[4], John S. Davies [5], Valarie A. Barr [6], Tomoki Kawai[7], Ottavia M. Delmonte[7], Marita Bosticardo [7], Mary Garofalo[8], Magda Carneiro-Sampaio[9], Raz Somech[10,11,12], Mohammad Gharagozlou[13], Nima Parvaneh [13], Lawrence E. Samelson[6], Thomas A. Fleisher[1], Anne Puel[14,15,16], Luigi D. Notarangelo [7], Bertrand Boisson [14,15,16], Jean-Laurent Casanova[14,15,16,17,18], Beata Derfalvi[19] & Sergio D. Rosenzweig [1] ✉

We describe the first cases of germline biallelic null mutations in ARPC5, part of the Arp2/3 actin nucleator complex, in two unrelated patients presenting with recurrent and severe infections, early-onset autoimmunity, inflammation, and dysmorphisms. This defect compromises multiple cell lineages and functions, and when protein expression is reestablished in-vitro, the Arp2/3 complex conformation and functions are rescued. As part of the pathophysiological evaluation, we also show that interleukin (IL)−6 signaling is distinctively impacted in this syndrome. Disruption of IL-6 classical but not trans-signaling highlights their differential roles in the disease and offers perspectives for therapeutic molecular targets.

Novel monogenic defects that result in dysregulation of the actin cytoskeleton are increasingly recognized to underlie human inborn errors of immunity (IEI)[1,2]. Actin-related protein 2/3 complex subunit 5 (ARPC5) is part of the ubiquitously expressed Arp2/3 complex, an actin nucleator involved in multiple cellular processes including cell movement, vesicle trafficking, and receptor endocytosis. This complex is composed of seven subunits: actin-related protein 2 and 3 (Arp2 and Arp3) and actin-related protein complex subunits 1–5, two of which

[1]Immunology Service, Department of Laboratory Medicine, Clinical Center, National Institutes of Health, Bethesda, MD, USA. [2]Neutrophil Monitoring Laboratory, Applied/Developmental Research Directorate, Frederick National Laboratory for Cancer Research, Frederick, MD, USA. [3]Laboratory of Pathology, Center for Cancer Research, National Cancer Institute, National Institutes of Health, Bethesda, MD, USA. [4]Electron Microscopy Laboratory, Biological Imaging Core, National Eye Institute, National Institutes of Health, Bethesda, MD, USA. [5]Predictive Toxicology Department of Safety Assessment, Genentech, South San Francisco, CA, USA. [6]Laboratory of Cellular and Molecular Biology, Center for Cancer Research, National Cancer Institute, National Institutes of Health, Bethesda, MD, USA. [7]Laboratory of Clinical Immunology and Microbiology, National Institute of Allergy and Infectious Diseases, National Institutes of Health, Bethesda, MD, USA. [8]Laboratory of Host Defenses, National Institute of Allergy and Infectious Diseases, National Institutes of Health, Bethesda, MD, USA. [9]Children's Hospital, Hospital das Clínicas da Faculdade de Medicina da Universidade de São Paulo (HC-FMUSP), São Paulo, Brazil. [10]Pediatric Department A and Immunology Service, Edmond and Lily Safra Children's Hospital, Tel Hashomer, Israel. [11]The Jeffrey Modell Foundation Israeli Network for Primary Immunodeficiency, New York, NY, USA. [12]Sheba Medical Center, Affiliated to the Sackler Faculty of Medicine, Tel Aviv University, Tel Aviv, Israel. [13]Division of Allergy and Clinical Immunology, Department of Pediatrics, Children's Medical Centre, University of Medical Sciences, Tehran, Iran. [14]St. Giles Laboratory of Human Genetics of Infectious Diseases, Rockefeller Branch, The Rockefeller University, New York, NY, USA. [15]Laboratory of Human Genetics of Infectious Diseases, Necker Branch, INSERM U1163, Necker Hospital for Sick Children, Paris, France. [16]Université Paris Cité, Imagine Institute, Paris, France. [17]Department of Pediatrics, Necker Hospital for Sick Children, AP-HP, Paris, France. [18]Howard Hughes Medical Institute, New York, NY, USA. [19]Department of Pediatrics, Division of Immunology, Dalhousie University and IWK Health Center, Halifax, NS, Canada. ✉e-mail: srosenzweig@cc.nih.gov

exist in 2 isoforms encoded by paralogous genes (ARPC1A, ARPC1B, ARPC2, ARPC3, ARPC4, ARPC5, and ARPC5L)[3,4]. Germline mutations disrupting several proteins that interact with the Arp2/3 complex are known causes of IEI, including Wiskott-Aldrich syndrome protein (WASP)[5,6]. So far, only *ARPC1B* mutations affecting the Arp2/3 complex have been described to cause a combined immunodeficiency with bacterial and viral infections, vasculitis, eczema, allergies, auto-immunity, bleeding tendency, and failure to thrive[7–11]. Herein, we report germline biallelic null mutations in *ARPC5* generating actin dysfunction and cytokine dysregulation, particularly interleukin (IL) −6-mediated signaling, and presenting with a complex phenotype of increased susceptibility to infections, autoimmunity, inflammation, and dysmorphisms.

## Results

### Case reports

Patient 1 (P1) was the eldest daughter born to healthy consanguineous parents of Lebanese descent (Fig. 1a–c and 2). She was born at 41 weeks gestation with intrauterine growth restriction. P1 became symptomatic within the first months of life with gastrointestinal and respiratory tract bleeding, hepatosplenomegaly, Evans syndrome (autoimmune hemolytic anemia and thrombocytopenia) and neu-trophilia. Her clinical course was remarkable for respiratory tract complications including recurrent/severe viral, bacterial, and fungal infections, pulmonary alveolar proteinosis and pneumatoceles; eczema; juvenile dermatomyositis-like focal myositis; delayed wound healing; recurrent episodes of paralytic ileus; celiac disease; hepatitis;

**a**

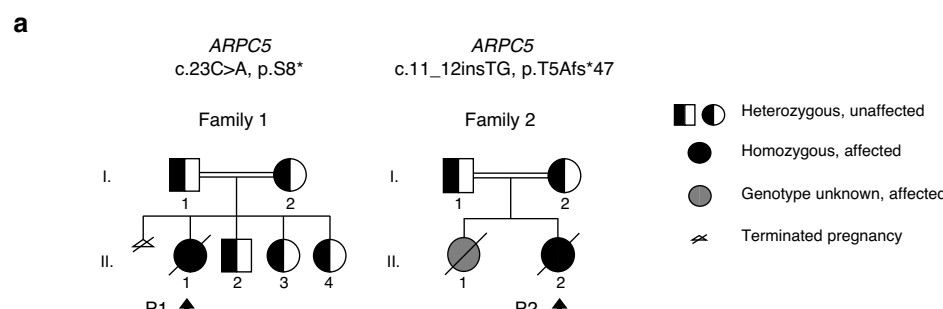

| | Patient 1, 15 years | Patient 2, 1 year |
|---|---|---|
| Age of onset | 1st months of life | 1st months of life |
| Infections | +++ | +++ |
| Impaired wound healing | +++ | +++ |
| Scoliosis | +++ | +++ |
| Pneumatoceles | +++ | +++ |
| Autoimmunity | +++ | - |
| Anemia | +++ | ++ |
| Leukocytosis | + | +++ |
| Lymphocytosis | + | + |
| Neutrophilia | + | +++ |
| Eosinophilia | +/- | +/- |
| Monocytosis | ++ | +++ |
| Thrombocytopenia | + | - |
| Elevated IgA (serum) | +++ | ++ |
| Elevated IgE (serum) | - | + |
| High acute phase reactants | +++ | +++ |

**c**

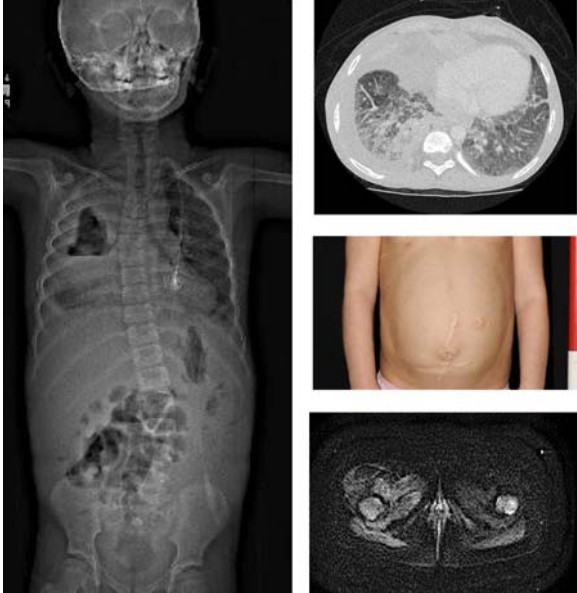

**Fig. 1 | ARPC5 deficiency – description and phenotype. a** Family pedigrees from patients with *ARPC5* variants. Arrows point to the patients studied (P1 in Family 1, P2 in Family 2); Roman numerals indicate generations. **b** Table depicting the main clinical and laboratory findings in P1 and P2. Negative symbols (−) denote absence; crosses, from (+) to (+++) indicate less to more severe phenotypes, respectively. **c** Patient 1 images. In the left image, a standing radiograph of P1 shows biconvex thoracolumbar scoliosis with convex right thoracic spinal curvature and convex left thoracolumbar spinal curvature; a right upper lobe pneumatocele is also identified. In the right images (upper, middle and lower, respectively), lung ground glass opacities, multiple abdominal scars product of abnormal wound healing, and right sided myositis of the thigh, are also detected.

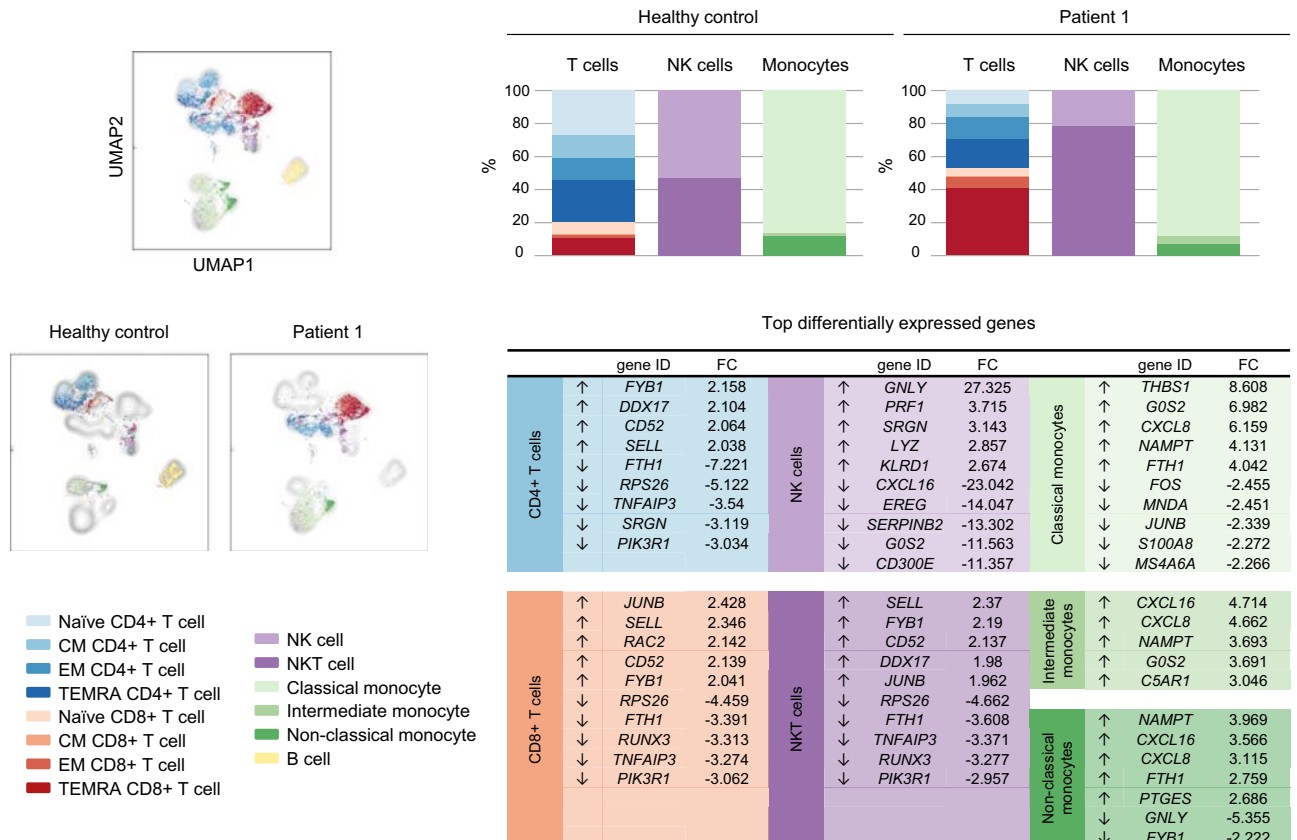

**Fig. 2 | Single-cell analysis data of peripheral blood mononuclear cells (PBMC) from patient 1 (P1) and a healthy control (HC).** Plots on the left show uniform manifold approximation and projection (UMAP) visualization of PBMC clusters. B lymphocytes were excluded from further analysis as P1 had been treated with anti-CD20 prior to sample collection. Bar graphs show the distribution of PBMC subpopulations in each sample, defined by surface protein markers using the BD Rhapsody platform. Top up- and down-regulated genes in PBMC subsets of P1 relative to HC are shown in the table. CM central memory, EM effector memory, FC expression fold change, NK natural killer, TEMRA terminally differentiated effector memory T cells re-expressing CD45RA.

minimal change disease nephropathy; short stature, facial dysmorphisms and scoliosis. Multiple autoantibodies were detected at various timepoints of her life; acute phase reactants, particularly C-reactive protein (CRP), were persistently elevated. Immunosuppressive and immunomodulatory treatments (e.g., steroids, rituximab, and high dose intravenous immunoglobulins) were used with partial and unsustained success. At 15 years of age, she died after a sudden episode of hemoptysis and hematochezia; an autopsy was not performed. Biospecimens from P1 were available for research studies.

Patient 2 (P2) was the youngest daughter born to healthy consanguineous Iranian parents (Fig. 1a, b). She became symptomatic since early in life when she presented with omphalitis. Her clinical course was remarkable for respiratory tract complications including severe/recurrent bacterial infections and pneumatoceles as well as skin infections, poor wound healing, multifocal aseptic bone lesions, and scoliosis. Neurodevelopmental delay, spasticity and brain atrophy were also evidenced. Acute phase reactants were persistently elevated. At the age of one year, she died because of progressive neurologic and respiratory disease; an autopsy was not performed. Cells from P2 were not available for research studies. P2's elder sister presented with severe, invasive bacterial infections (i.e., *Staphylococcus aureus* and *Pseudomonas aeruginosa*) since the neonatal period; hepatosplenomegaly and hematochezia were also present, but no neurologic manifestations were observed. The patient died at six months of age due to progressive perineal necrotic infectious complications; an autopsy was not performed. Biospecimens from this patient were not available for testing.

**P1 and P2 detailed clinical descriptions are available as a Supplementary Note**

Clinical laboratory evaluation on P1 and P2 revealed anemia and elevated white blood cell counts, including neutrophilia, and lympho-monocytosis (Fig. 1b, Supplementary Table 1). P1's extended analysis showed inverted CD4/CD8 ratios, with increased number of terminally differentiated CD8+ T cells and decreased number of T regulatory cells (Fig. 2, Supplementary Table 2 and Supplementary Fig. 1). B cells (prior to anti-CD20 treatment) were increased, with high frequency of age-associated B cells, a subset linked to autoimmunity[12] (Supplementary Table 2 and Supplementary Fig. 2). NK cells fell within normal ranges, but NKT cells were elevated. These mononuclear cell changes were complemented by transcriptional/functional defects as determined by single cell differentially expressed gene analysis (Fig. 2, Supplementary Table 2). T-cell proliferation was normal in response to phytohemagglutinin and T-cell receptor (TCR) stimulation (Supplementary Fig. 1). Specific antibody responses to recall protein were normal but unsustained to polysaccharide antigens (Supplementary Table 1). B cells proliferated normally upon various stimulating conditions (Supplementary Fig. 2). IgG, IgA, and IgM serum levels were normal/high in P1 and P2; IgE was elevated in P2 (Supplementary Table 1).

**Candidate variant selection**

Whole exome sequencing trio analysis was performed in two families with unknown genetic diseases and phenotypic similarities (Fig. 1a–c). Among the candidate variants (Supplementary Table 3), a

homozygous nonsense variant in *ARPC5* (NM_005717 c.23C>A p.S8*) in P1 and a homozygous frameshift variant in *ARPC5* (NM_005717 c.11_12insTG, p.T5Afs*47) in P2 were Sanger confirmed and prioritized. These variants had Combined Annotation Dependent Depletion (CADD) scores of 38 and 24.4, respectively, and were not found in the Genome Aggregation Database (gnomAD v2.1.1) (Supplementary Fig. 4)[13]. *ARPC5* encodes ARPC5, a component of the Arp2/3 complex. A homozygous missense variant in *BRINP2* (NM_021165 c.184C>T, p.R62W; CADD score of 30; not found in gnomAD) was also detected in P2, and possibly associated with her neurologic manifestations.

## Mutant ARPC5 biochemical and functional studies

While the mutant *ARPC5* mRNA was detected (Supplementary Fig. 5), expression of ARPC5 was not detected in peripheral blood mononuclear cells (PBMC), T-cell blasts, and fibroblasts from P1 (Fig. 3a, Supplementary Fig. 6). N-terminus truncated proteins, resulting from alternative transcription initiation sites were not detected either. The heterozygous parents showed intermediate expression levels of ARPC5. Immunoblotting of the other Arp2/3 complex subunits revealed reduced expression of ARPC1A and ARPC1B in P1 (Fig. 3a, Supplementary Fig. 6). ARPC5 expression was found to be normal in two patients with ARPC1B deficiency (Supplementary Fig. 6). Formation of Arp2/3 complexes was quantitatively and qualitatively impaired in P1. Arp2/3 complexes containing ARPC5 were not present in P1 but were detected in HC fibroblasts lysates. Arp2/3 complexes containing ARPC2 were present in P1 at lower expression levels and with higher electrophoretic mobility relative to HC (Fig. 3b).

Immunofluorescence microscopy evaluation of P1's early adherent fibroblasts revealed a high number of filopodia (elongated, finger-like actin protrusions), whereas HC cells predominantly formed lamellipodia (flat, sheet-like actin projections) (Fig. 4a). Continuous impedance monitoring of P1's fibroblasts showed lower spreading speed compared to HC's cells (Fig. 4a). Treatment of HC's fibroblasts with the Arp2/3 inhibitor CK-666 resulted in reduced spreading capacity mimicking P1's fibroblasts behavior (Supplementary Fig. 7). A similar pattern of aberrant spreading characterized by enrichment of filopodia and absence of lamellipodia was observed in P1's neutrophils (Fig. 4b, Supplementary Fig. 8, and Supplementary Movie 1). Real-time sequential imaging revealed that P1's cells covered a smaller area over time compared to HC's neutrophils (Fig. 4b). Cell motility was assessed in P1's fibroblasts via a time course wound healing assay. P1's fibroblasts were slower than HC's cells to close the in-vitro wound gap (Fig. 4c). Migration of neutrophils was assessed in response to N-formylmethionyl-leucyl-phenylalanine (fMLF). Directed migration was severely impaired in P1's neutrophils, as no cells were able to complete migration during the observation period, unlike HC's cells (Fig. 4d and Supplementary Movie 2).

## Wild-type ARPC5 expression rescued abnormal findings

Wild-type (WT) ARPC5 expression was rescued in P1's fibroblasts via plasmid transfection or lentiviral transduction. Expression of WT ARPC5 restored ARPC5 protein expression and increased ARPC1A and ARPC1B expression levels compared to empty vector-transfected P1's cells (Fig. 5a, Supplementary Fig. 9). Expression of WT ARPC5 also recovered expression of ARPC5- and ARPC2-containing Arp2/3 complexes, which exhibited similar electrophoretic mobility to HC (Fig. 5b).

ARPC5 staining localized to the nucleus, to scattered dots throughout the cytoplasm, and to the edge of the plasma membrane on rescued cells, compatible with expected ARPC5 subcellular localization[14–16] (Fig. 5c, Supplementary Fig. 9). The adhesion and spreading pattern measured by impedance monitoring improved in wild-type ARPC5 expressing cells (Fig. 5d). Restored lamellipodia formation and wound healing could also be detected in ARPC5-rescued cells (Fig. 5c, e, and Supplementary Fig. 9). Altogether, these findings confirmed disease causality for *ARPC5* mutations.

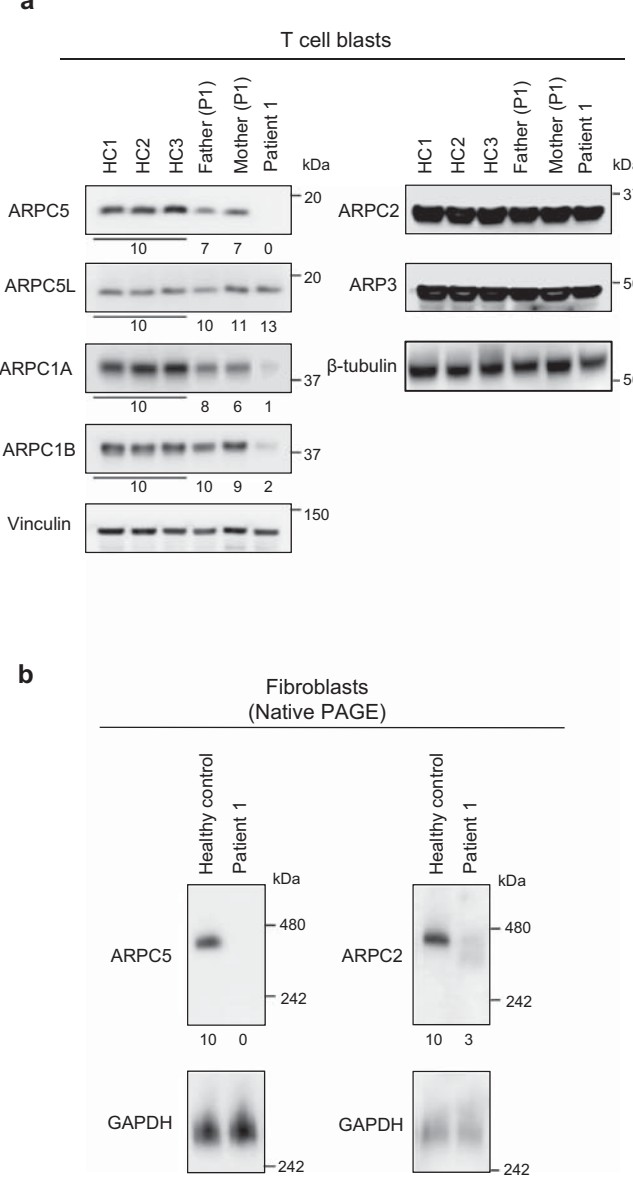

**Fig. 3 | Mutant ARPC5 biochemical studies. a** Western blotting analysis of patient 1 (P1), her parents, and healthy controls (HC) T-cell blasts lysates with antibodies specific to Arp2/3 complex subunits, as indicated. Vinculin and β-tubulin were used as loading controls. **b** Western blotting analysis of cell extracts from P1's and HC's fibroblasts after native gel electrophoresis. ARPC5- or ARPC2-specific antibodies were used to probe the Arp2/3 complex; GAPDH was used as a loading control. Blots in this figure are representative of at least two independent experiments. The numbers below the western blotting images (**a** and **b**) represent protein expression levels, quantitatively measured in relation to healthy controls, after normalization to the loading control. Healthy controls' average value was set at 10. PAGE polyacrylamide gel electrophoresis. Source data are provided as a Source Data file.

## Clinical phenotype, IL-6 signaling and molecular targeting

P1 (and to some extent P2, despite her young age) presented with clinical features resembling both *STAT3* dominant-negative (DN; Job's syndrome) and *STAT3* gain-of-function (GOF) mutations. While P1's facial dysmorphism, eczema, bacterial -*Staphylococcus aureus*- and fungal -*Candida* spp., *Aspergillus fumigatus*- pneumonias, poor wound healing, pneumatoceles, and scoliosis were suggestive of a Job's-like syndrome; her short stature, skin manifestations, autoimmune cytopenia -Coombs+ autoimmune hemolytic anemia, thrombocytopenia-, celiac disease, autoimmune endocrinopathies -hypothyroidism-, and

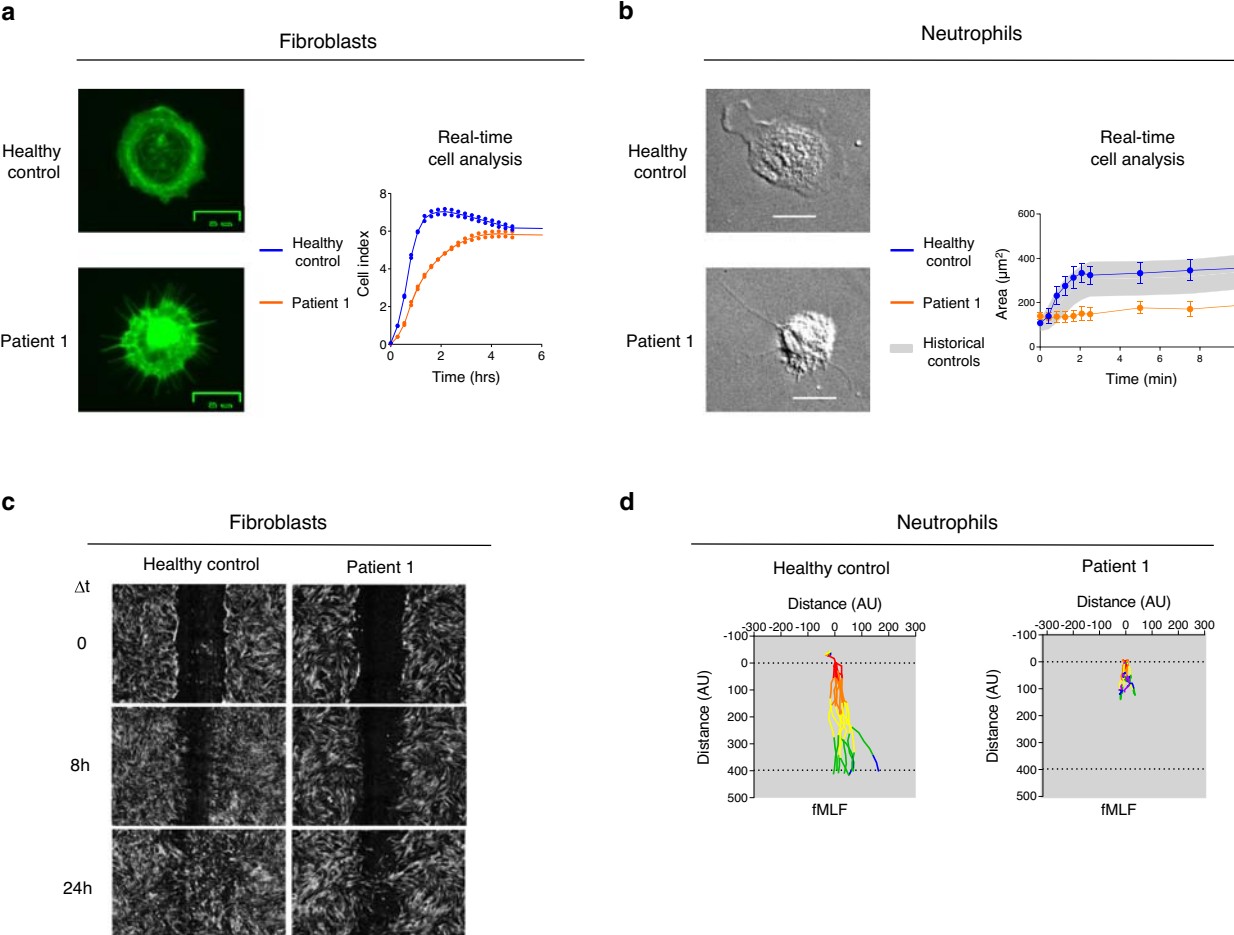

**Fig. 4 | Mutant ARPC5 functional studies. a** Fibroblasts from patient 1 (P1) and a healthy control (HC) were seeded on a retronectin-coated surface and fixed after 20 min. Actin protrusions were assessed via immunofluorescent staining with phalloidin. Adhesion and spreading of fibroblasts were monitored via real-time, impedance-based cell assay (graph). Impedance values are reported as cell index (CI). Curves represent the mean CI value with individual replicate data shown as dots. **b** Light microscopy images from P1 and HC neutrophils. The scale bars indicate 10 μm. The area over time covered by P1's neutrophils compared to HC is shown in the graph. Curves represent the mean (±standard deviation) area from 5 to 6 cells. **c** Wound healing assay conducted with P1's and HC's fibroblasts. The gap length shown at 0 h corresponds to 0.94 mm. **d** Neutrophil chemotaxis in response to N-formylmethionyl-leucyl-phenylalanine (fMLF). The cells were incubated for 1 h and images were captured every 30 s. Single cell migration traces of ten randomly selected cells are shown. The position of each cell was anchored at the origin at t = 0. All results in this figure are representative of at least two independent experiments. Source data are provided as a Source Data file.

autoimmune hepatitis were suggestive of a STAT3 GOF-like phenotype. When her cytokine responses to IL-2, IL-4, IL-7, IL-21, interferon (IFN)α and IFNγ were evaluated in T cells and monocytes, no abnormalities were detected (Supplementary Fig. 10). Other T-cell functions such as TCR rearrangement, signaling, and microcluster formation were normal in P1's T-cell blasts (Supplementary Figs. 11–14). TCR signaling and migration were also normal in CRISPR/Cas9 generated *ARPC5*-knockout Jurkat cells (Supplementary Fig. 15). However, signaling through IL-6, primarily dependent on STAT3 and to a lesser extent on STAT1, was markedly diminished. Phosphorylation of STAT3 in response to in-vitro IL-6 stimulation was nearly absent in P1's CD4+ T cells yet was present in response to IL-21 (Fig. 6a). When the expression of cell surface IL-6 receptor complex (IL-6RC) components IL-6Rα and gp130 were evaluated, very low levels of IL-6Rα were detected, while gp130 expression was mildly reduced (Fig. 6b). When IL-6RC signaling was tested in P1's fibroblasts, response to IL-6 was also nearly absent, while gp130-dependent/IL-6Rα-independent responses to oncostatin-M and IL-11 were fully preserved (Fig. 6c). Treatment of healthy controls' CD4+ T cells with CK-666 resulted in a reduction of ~21% in the surface expression of IL-6Rα, and a 1.4-fold increase in soluble IL-6Rα when compared to the vehicle treated samples (Fig. 6d), suggesting that the actin defect was indeed contributing to the impaired IL-6Rα surface expression levels. The levels of transcription for both membrane-bound IL-6Rα and soluble IL-6Rα were not affected by treatment with CK-666 (Supplementary Fig. 16). As P1 had consistently elevated CRP levels, a known IL-6-dependent readout[17], we hypothesized that trans-signaling, an alternative model to IL-6 classical signaling reliant on the soluble form of IL-6Rα[18], was likely enhanced. While a clinical serum cytokine panel showed modest elevation of IL-6 (29.48 pg/mL, within 85–97.5% of normal distribution), we found markedly increased levels of both soluble IL-6Rα (sIL-6Rα) and sIL-6Rα complexed with IL-6 in P1's plasma (Fig. 7a). When tested in-vitro, P1's CD4+ T cells and fibroblasts were able to phosphorylate STAT3 in response to an IL-6/IL-6Rα fusion protein, known as hyper-IL6[19] (Figs. 6c and 7b). Such response could be controlled by pre-treatment of cells with anti-IL-6Rα (tocilizumab) or soluble gp130Fc[20] (Fig. 7b). P1 died before in vivo IL-6 modulation could be explored.

## Discussion

Herein we describe two unrelated patients with a previously undescribed IEI associated with germline biallelic null mutations in *ARPC5*. The patients presented with recurrent and severe infections, early-onset autoimmunity, inflammation, and dysmorphisms. In addition, we show how ARPC5-dependent actin disruption can impact IL-6

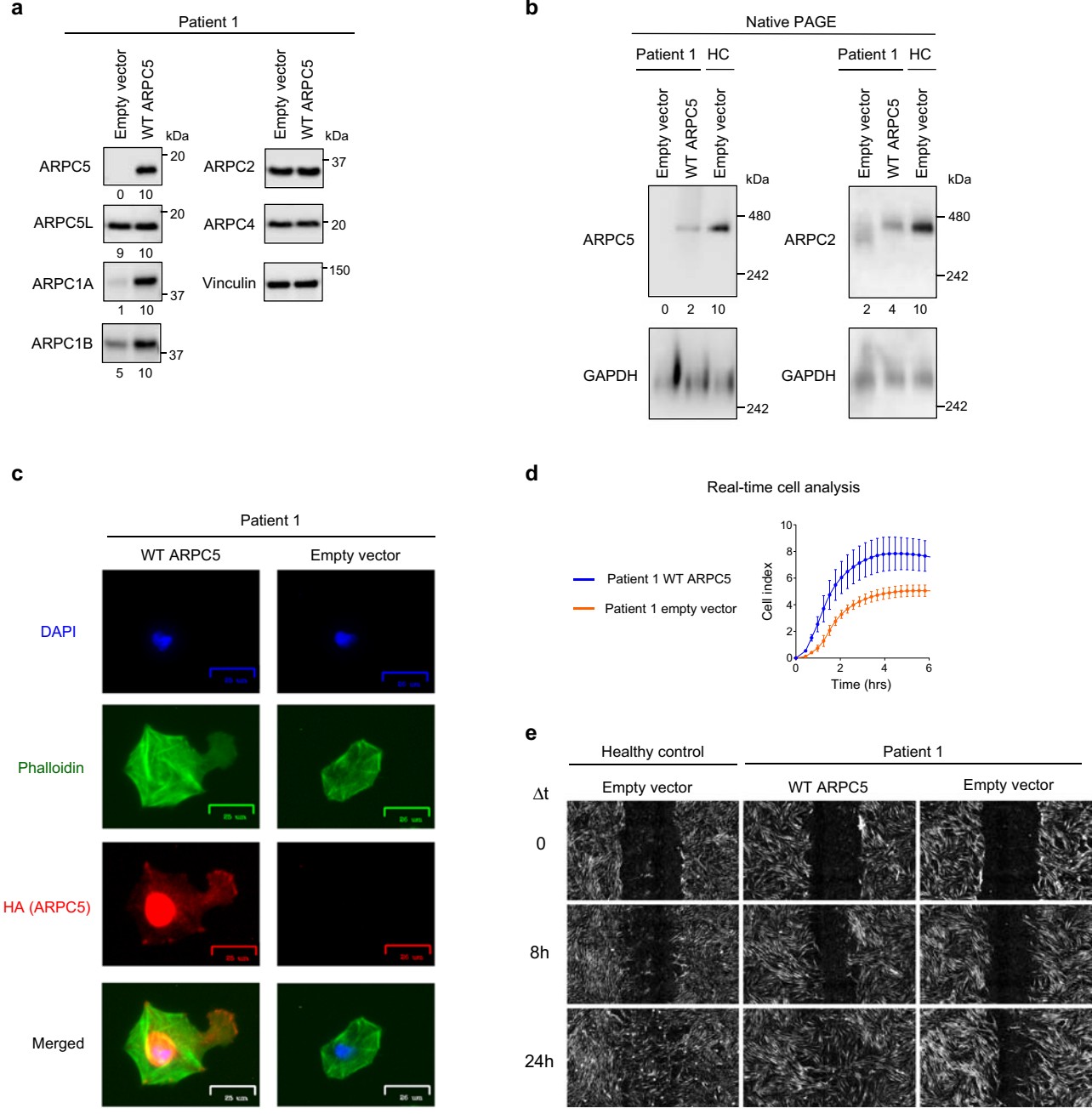

**Fig. 5 | Rescue of defective functions by expression of wild-type ARPC5.**
**a** Protein expression by immunoblotting of individual Arp2/3 complex subunits in lysates from P1's fibroblasts transiently transfected (efficiency 20–40%) with empty vector or a plasmid encoding wild-type (WT) *ARPC5*. Vinculin was used as a loading control. **b** Western blotting analysis after native gel electrophoresis of cell extracts from P1's and healthy control's (HC) fibroblasts expressing WT ARPC5 by lentiviral transduction. ARPC5- or ARPC2-specific antibodies were used to probe the Arp2/3 complex. GAPDH was used as a loading control. **c** Immunofluorescence images of representative fibroblasts from P1 transfected with WT *ARPC5* (left column) versus cells transfected with empty vector (right column). Cells were seeded on a retronectin-coated surface and fixed after 120 min. **d** Real-time, impedance-based monitoring of P1's *ARPC5*-rescued fibroblasts versus *ARPC5* mock-transduced cells. Impedance values are reported as cell index (CI). Curves represent the mean (±standard deviation) cell index value from four technical replicates. **e** Wound healing assay with WT *ARPC5*-rescued fibroblasts from P1 (middle column) and mock-transduced P1 cells (right column), HC cells were used as controls (left column). The gap length shown at 0 h corresponds to 0.94 mm. All results in this figure are representative of at least two independent experiments. The numbers below the western blotting images represent protein expression levels relative to WT *ARPC5*-rescued P1 fibroblasts (**a**) or HC fibroblasts (**b**), after normalization to the loading control. PAGE polyacrylamide gel electrophoresis. Source data are provided as a Source Data file.

signaling, that can be considered as a therapeutic molecular target for this disease.

Genetically, ARPC5 deficiency is inherited in an autosomal recessive manner with no evidence of haploinsufficiency (i.e., heterozygous parents were asymptomatic); while clinical penetrance seems complete (i.e., both individuals with biallelic mutations were symptomatic),

more patients will have to be described to fully confirm disease penetrance and expressivity.

We demonstrated that ARPC5 expression was lost in patient cells, resulting in lower expression of interacting proteins ARPC1B and ARPC1A, but not of other Arp2/3 complex components. The dependence of ARPC1 isoforms on ARPC5 levels has been previously shown

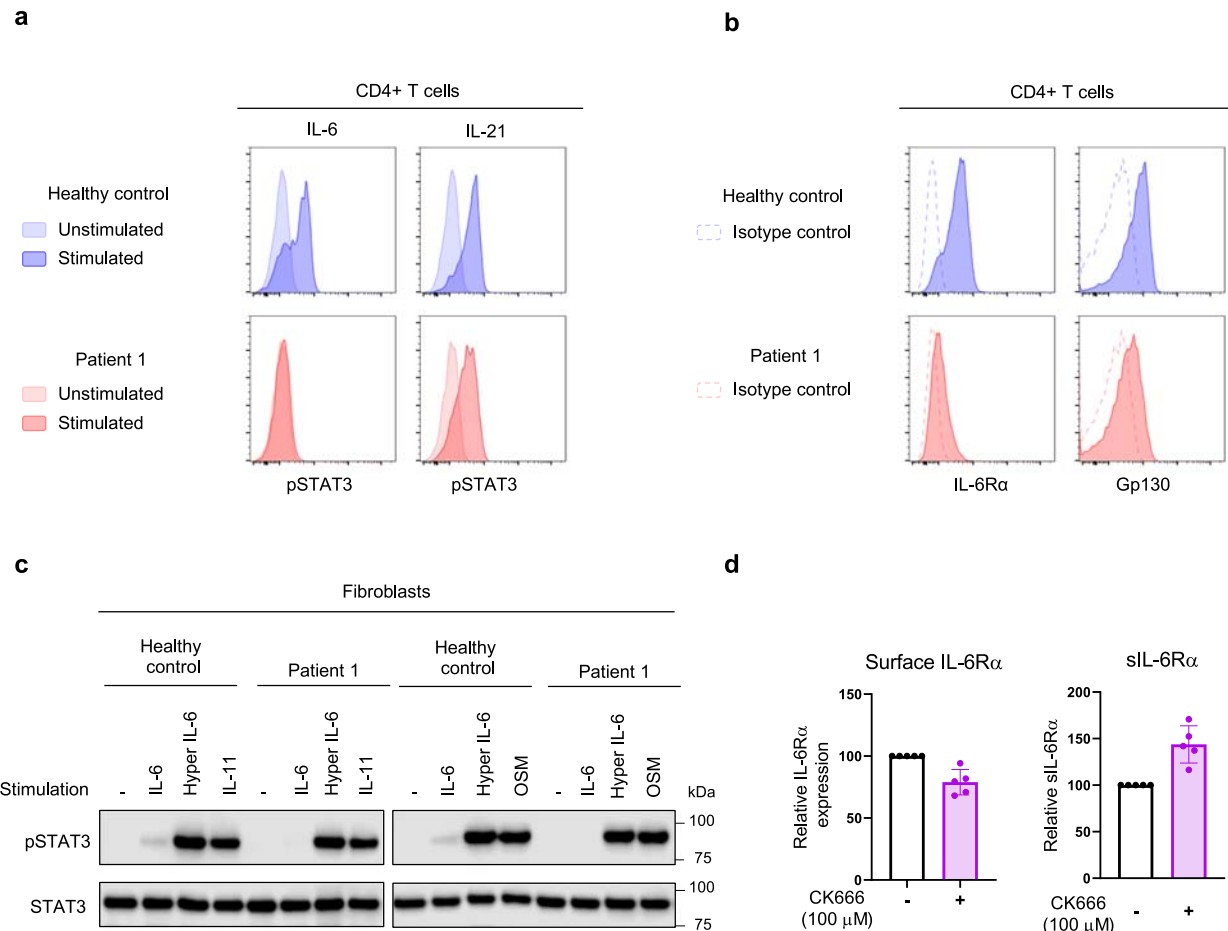

**Fig. 6 | Interleukin-6 signaling. a** Histograms showing signal transducer and activator of transcription 3 (STAT3) phosphorylation by flow cytometric analysis of P1's and healthy control's (HC) CD4+ T cells after short-term in-vitro cytokine stimulation. Data are representative from three independent experiments. **b** Expression of the cell surface IL-6 receptor complex components (IL-6Rα and gp130) in P1's and HC's CD4+ T cells, by flow cytometry. Data are representative from three independent experiments. **c** STAT3 signaling in P1 and HC evaluated by immunoblotting of fibroblasts lysates in response to short-term in-vitro stimulation with gp130-family cytokines (IL-6, IL-11, and oncostatin-M [OSM]) and hyper IL-6.

Data are representative from two independent experiments. **d** Enriched CD4 T cells were treated either with a vehicle (DMSO) or Arp2/3 complex inhibitor CK-666 (100 mM) for 22 h. Flow analysis was used to evaluate the surface expression of IL-6Rα. The supernatants of the same samples treated with CK-666 were collected, and the Luminex assay was utilized to determine the concentration of soluble IL-6Rα. The IL-6Rα expression levels were calculated relative to the vehicle treated controls. Data are expressed as mean + standard deviation (SD) from five different healthy controls. Source data are provided as a Source Data file.

in HeLa cells[21] and is now confirmed in human disease. In contrast, ARPC1B deficiency does not seem to affect ARPC5 expression levels, suggesting that ARPC5 deficiency might have a broader impact on biology and disease. Moreover, depletion of ARPC1B levels in the patient with ARPC5 deficiency supports the phenotypic overlaps between the two diseases[7–11].

Functionally, the Arp2/3 complex is recognized by its unique ability to nucleate actin filaments at an angle from a preexisting filament, resulting in a branched network of polymerized actin[3]. Branched actin is the predominant form of actin organization in lamellipodia, podosomes and invadopodia, all required but not indispensable for cell motility[22]. Experimental disruption of the Arp2/3 complex results in abrogation of lamellipodial structures with enrichment of filopodia, as confirmed in P1's fibroblasts and neutrophils[23,24]. Similarly defective actin distribution patterns have been documented in platelets, neutrophils, and T cells from patients with ARPC1B deficiency[7,8,10], as well as in patients with HEM-1 deficiency[6], a protein required for the Arp2/3 complex to generate lamellipodia[25].

We showed that fibroblast migration was impaired in ARPC5 deficiency. This finding correlated with the poor and delayed wound healing experienced by our patients. Impaired wound

healing was also observed with in-vitro testing of murine $Arpc3^{-/-}$ fibroblasts[24] and $Arpc5$-silenced rat smooth muscle cells[26]. Interestingly, migration of fibroblasts from a patient with ARPC1B deficiency was unaffected[8], suggesting that not all Arp2/3-related defects behave as phenocopies in the same cell lineage and function. Moreover, while fibroblast defects are certainly contributing to the wound healing defect, they are unlikely to be the only cause: the hyperinflammatory state and impaired neutrophil migration, both seen in other PID/IEI with impaired wound healing, must be considered as other likely responsible factors[27,28].

Thrombocytopenia in ARPC5 deficiency is probably another multifactorial complication in this disease. Despite peripheral blood thrombocytopenia, bone marrow biopsies on P1 never showed increased megakaryocytes but did present with dysplastic features, similarly to what was reported in patients with ARPC1B deficiency[7], which is also markedly diminished in ARPC5 deficiency. Her persistent hepatosplenomegaly and broad autoimmunity profile may have also potentially contributed to platelet sequestration and thrombocytopenia.

Although quantitatively decreased and qualitatively altered, the assembly of Arp2/3 complexes was not completely abrogated

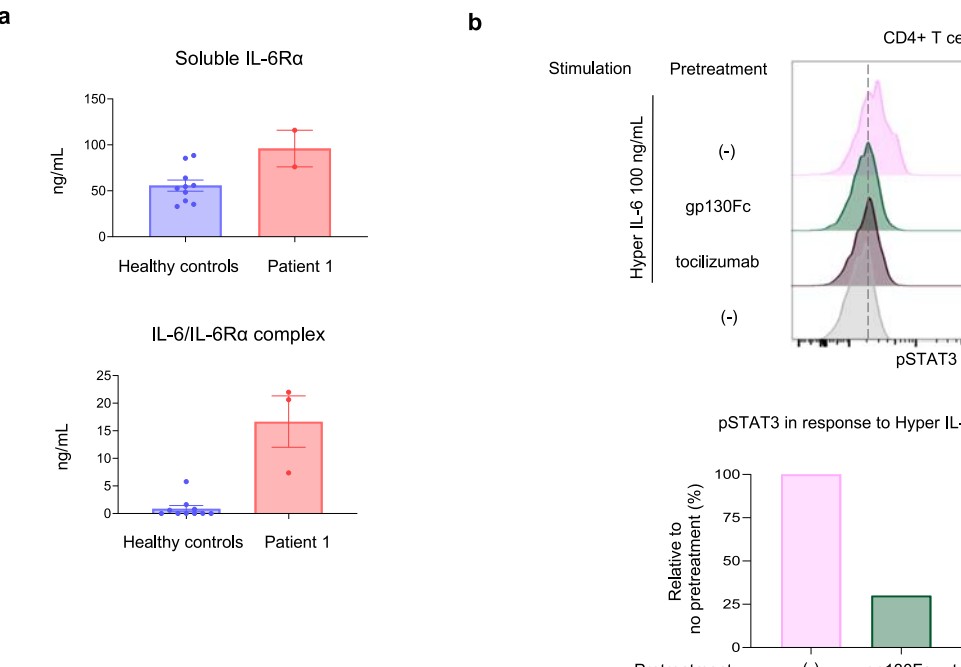

**Fig. 7 | Interleukin-6 molecular targeting. a** Plasma levels of soluble IL-6Rα measured by Luminex assay (top panel) and soluble complexed IL-6/IL-6Rα measured by ELISA (bottom panel) in patient 1 (P1) and five healthy controls (HC). Results are shown as mean + SEM. Data generated from two independent experiments, performed in duplicate each time. **b** IL-6 trans-signaling modulation in P1's CD4+ T cells. P1's cells were pretreated with soluble gp130Fc (1 μg/mL) or tocilizumab (10 μg/mL) for 90 min. A subset of cells was left untreated to serve as control. Then, cells were stimulated with hyper IL-6 (100 ng/mL) for 20 min. A subset of cells was not stimulated to serve as baseline. Sequentially, phosphorylation of STAT3 was measured by flow cytometry (upper panel). The bar graph shows soluble gp130Fc and tocilizumab pretreatment effect on STAT3 phosphorylation in response to hyper IL-6 stimulation compared to untreated (−), stimulated cells (lower panel). This experiment was conducted once. Source data are provided as a Source Data file.

in the context of ARPC5 deficiency. The same was seen in platelet lysates from patients with ARPC1B deficiency[7] and likely contribute to the non-embryonic lethality of these genotypes[22]. Whether Arp2/3 complex integrity in these scenarios is maintained by alternative isoforms of ARPC1 (as reported by Leung et al.)[29] and ARPC5, or by non-canonical conformations of the complex lacking these subunits[30], is yet to be established. Of note, different compositions of the Arp2/3 complex have been shown to nucleate actin with distinct efficiency and to present variable stability. Specific combinations of the complex are likely suited to particular physiologic processes[21,30]. In this setting, Faessler et al., very recently reported how murine and rat ArpC5 subunit isoforms (i.e., Arpc5 and Arpc5L) can alternatively and differently regulate particular Arp2/3-dependant functions. Interestingly, some differences became evident between human vs. rodent species as ARPC5 deficiency in our work does not seem to affect ARPC5L accumulation, while ArpC5 deficiency consistently increased the ArpC5L levels likely affecting the functions dependent on each type of Arp2/3 complexes[16]. Not surprisingly, the data presented here showed that depriving cells of the ARPC5 subunit resulted in dysregulation of several but not all Arp2/3-dependent functions. Altogether, these features help to explain why signaling through IL-6Rα but not all surface-expressed cytokine receptors are affected by ARPC5 deficiency. Interestingly, in depth analysis of the impact of ARPC5 deficiency on other T-cell functions failed to detect deleterious consequences as previously demonstrated in ARPC1B deficiency[10,31].

Despite the inherent limitations of studying rare diseases with small number of patients[32], we showed that two unrelated patients carrying biallelic null mutations in *ARPC5* suffered from a previously unreported disease. While the association between ARPC5 deficiency and CNS or other syndromic manifestations will be further clarified as more patients are described, we efficiently proved that the ARPC5 protein was not expressed, Arp2/3 complex formation was impaired, and subsequently different actin-related functions were variably affected in specific cell lineages in this disease. Rescue of ARPC5 expression reestablished Arp2/3 complex conformation and function. As part of the clinical-pathophysiological evaluation, we also showed that the IL-6 pathway was distinctively impacted in this disease. IL-6 signaling was tested, trans-signaling seemed to have a critical role in the disease and offered a molecular target for treatment. This work emphasizes the importance of studying rare genetic diseases through rigorous methodological criteria to unveil the underlying pathophysiology mechanism and potential therapeutic opportunities.

## Methods
### Patients and samples
The patients' legal representatives provided informed consent at their respective institutions, in accordance with the Declaration of Helsinki, under protocols approved by local institutional review boards at the Rockefeller University, and/or National Institutes of Health (NIH)/ National Institute of Allergy and Infectious Diseases (NIAID) protocol 10-I-0216; ClinicalTrials.gov Identifier: NCT01222741. The patients' legal representatives consented to the publication of medical information and images related to the cases. Samples from Patient 1's family were collected at several time points between 2019 and 2021 at Dalhousie University, Halifax, Nova Scotia, Canada and sent to NIH for genetic and functional studies; Patient 2's family samples were collected at different timepoints between 2012 and 2013 at the Children's Medical Center, Tehran, Iran and sent to the Rockefeller University for genetic testing. Blood samples and biopsies from patients, family members, and healthy donors were obtained following standard of care under approved protocols by the National Institutes of Health institutional review board.

## Whole exome sequencing (WES) and filtering of candidate variants

WES was performed using genomic DNA extracted from PBMCs using the Illumina–IDT Exome Enrichment Kit and HiSeq 2500 instrument according to the manufacturer's protocols. To prioritize genetic variants, we filtered the results to include only novel or rare and potentially functional non-synonymous, nonsense, indel, intronic and splicing variants, using tools in the Ensembl Variant Effect Predictor (VEP; https://useast.ensembl.org/info/docs/tools/vep/index.html) and ANNOVAR package (annovar.openbioinformatics.org), including gene/amino acid annotation (refGene), allele frequencies (gnomAD, NCBI dbSNP), functional prediction scores (SIFT, PolyPhen2, LRT, MutationAssessor, MutationTaster,PROVEAN, MetaSVM, MetaLR,RE-VEL, CADD, DANN, FATHMM, fathmm-MKL, GenoCanyon), conservation scores (GERP++, SiPhy), and genotype/phenotype correlation databases (InterVar, ClinVar, COSMIC). Variants in segmental duplication regions were excluded. We used Genedistiller 2 (http://www.genedistiller.org/) to identify genes associated with immune response. Finally, variants were finally prioritized based on clinical correlation.

## Sanger sequencing

Selected next-generation sequencing results were confirmed and carrier testing was performed by Sanger sequencing. Genomic DNA was PCR-amplified using M13-tagged specific variant-flanking primers, and the PCR products were subjected to Sanger sequencing using M13-primers and the BigDye Terminator v1.1 Cycle Sequencing Kit (Applied Biosystems, Cat. 4337452) according to the manufacturer's protocol. The Sanger sequencing data was analyzed using DNAStar Lasergene 16 SeqMan.

## Single cell RNA sequencing (scRNA-Seq)

We performed a targeted scRNA-seq approach using the BD Rhapsody Single Cell Analysis Systems (BD Biosciences). PBMC from a patient with ARPC5 deficiency (P1) and a healthy control (HC) were labeled with sample tags (BD Human Single Cell Multiplexing Kit) and AbSeq antibodies (CD1c, CD3, CD4, CD5, CD8, CD11c, CD14, CD16, CD18, CD19, CD20, CD21, CD23, CD24, CD25, CD27, CD28, CD31, CD38, CD40, CD44, CD45RA, CD45RO, CD56, CD62L, CD123, CD127, CD134, CD137, CD161, CD183, CD185, CD186, CD196, CD197, CD272, CD278, CD279, CD303, CD314, CD337, CD357, CD366, HLA-DR, IgD, IgM, TCRαβ, TCRγδ, Vα24-Jα18) and counted. Single-cell capture was performed using the BD Rhapsody Express following manufacturer's instructions. The libraries were pooled and sequenced on NovaSeq 6000 S2 as a dual index pair-end run. Fastq files were processed using BD's Rhapsody analysis pipeline on the Seven Bridges Platform using default parameters, the GRCh38-PhiX-gencodev29 reference genome, and the gencodev29-20181205 transcriptome annotation. Processed data (molecules per gene per cell based on DBEC error correction) were analyzed using BD SeqGeq v1.8 and the Seurat plug-in to perform dimensionality reduction (UMAP) and differential analysis output for selected cell subsets (CD4+ T cells, CD8+ T cells, NK cells, NKT cells, classical monocytes [CD14+, CD16−], intermediate monocytes [CD14+, CD16+], and non-classical monocytes [CD14dim, CD16+]). The differential analysis expression ratios were uploaded to the Ingenuity Pathway Analysis server (QIAGEN, IPA) and converted to expression fold changes. To identify clusters of diseases or biological functions that are predicted to increase or decrease similarly across different datasets, IPA core analyses of the fold changes for the subsets were compared using the Diseases and Functions Heat Map (filtered by z-score >2).

## Cell culture

PBMC were isolated from whole blood via density gradient centrifugation using Ficoll-Paque Plus (GE Healthcare, Cat. 17144002). PBMC and Jurkat T cells (Clone E6-1, ATCC, Cat. TIB-152) were cultured in RPMI-1640 medium (Gibco, Cat. 61870036) with 10% fetal bovine serum (FBS), 2 mM L-glutamine, 100 U/mL penicillin, and 100 μg/mL streptomycin at 37 °C in a humidified 5% $CO_2$ incubator. T-cell blasts were generated via stimulation of PBMC with soluble anti-CD3 (1 μg/mL, Invitrogen, Cat. 16003785) and anti-CD28 (1 μg/mL, Invitrogen, Cat. 16028985) antibodies plus IL-2 (10 ng/mL, Peprotech, Cat. 20002). IL-2 was added every 2–3 days and cells were harvested after 7–10 days. Raji B cells were grown in IMDM media (Gibco, Cat. 31980030) with 10% heat inactivated FBS, 1% Penicillin-Streptomycin, 1% NEAA (non-essential amino acid supplement) and 1% Glutamax. Neutrophils were isolated from heparinized blood by discontinuous gradient centrifugation using Ficoll-Paque Premium (GE Healthcare, Cat. 17544202). Contaminating red blood cells were removed by 3% dextran (Government Scientific Source, Inc.,551002508007, Pharmaceutical quality, MW 250,000) followed by two hypotonic lysis of 33 mM NaCl. Re-equilibration solution (267 mM NaCl) was added within 30 s of each lysis to normalize osmolarity[33]. Dermal fibroblasts from P1 and HC were obtained from skin punch biopsies following standard procedures. Immortalized fibroblasts cell lines were generated using a human telomerase reverse transcriptase (hTERT) cell immortalization kit (Alstem, Cat. CILV02) following manufacturer's protocol. Fibroblasts were cultured in DMEM medium supplemented with 10% FBS, 2 mM L-glutamine, 100 U/mL penicillin, and 100 μg/mL streptomycin (Gibco) at 37 °C in a humidified 5% $CO_2$ incubator. Cells were routinely tested for mycoplasma contamination.

## Western blotting

For Arp2/3 complex or individual Arp2/3 complex subunits protein expression studies, cells under resting condition were used. For signal transducer and activator of transcription 3 (STAT3) phosphorylation analysis, overnight serum-starved fibroblasts, under resting condition or stimulated for 15 min with either IL-6 (Peprotech, Cat. 20006), IL-6/IL-6R alpha protein chimera (Hyper IL-6, R&D Systems, Cat. 8954SR), IL-11 (Peprotech, Cat. 20011), or Oncostatin-M (Peprotech, Cat. 20010), all at 100 ng/mL, were used. For T-cell receptor (TCR) activation studies, 1 h serum-starved T-cell blasts, under resting condition or stimulated for 1 or 10 min with Dynabeads human T-activator CD3/CD28 (Gibco, Cat. 11131D) were used. Cell extracts were prepared in lysis buffer (50 mM Tris pH 7.4, 150 mM NaCl, 2 mM EDTA, 0.5% Triton X-100, and 0.5% NP40) with protease and phosphatase inhibitor cocktail (Sigma, Cat. PPC1010) on ice for 30 min. Samples were adjusted to have matching concentrations of total protein. Except for native Arp2/3 complex analyses, all other western blotting experiments were conducted under denaturing conditions in which cell lysates were boiled in the presence of NuPAGE LDS sample buffer (Invitrogen, Cat. NP0007) and NuPAGE sample reducing agent (Invitrogen, Cat. NP0004). Then, 5–10 μg of total protein from cell lysates were loaded into NuPAGE 4 to 12%, Bis-Tris, 1.0–1.5 mm, protein gels (Invitrogen, Cat. NP0323) for SDS-PAGE electrophoresis. Subsequently, proteins were transferred to a nitrocellulose membrane using the Trans-blot turbo transfer system (Bio-Rad, Cat. 1704150). For native Arp2/3 complex analyses, NativePAGE sample buffer (Invitrogen, Cat. BN2003) and NativePAGE 5% G-250 sample additive (Invitrogen, Cat. BN2004) were added to cell extracts, without heating, and 2 μg of total protein from cell extracts were loaded into NativePAGE 4 to 16%, Bis-Tris, 1.0 mm, protein gels (Invitrogen, Cat. BN2112BX10) for blue native PAGE electrophoresis following manufacturer's protocol. Subsequently, proteins were transferred to a PVDF membrane using the iBlot transfer system (Invitrogen). All membranes were blocked in 5% (wt/vol) of nonfat dry milk (Cell Signaling Technology, Cat. 9999) in 1X Tris-buffered saline with 0.1% Tween 20 (TBS-T, Cell Signaling Technology, Cat. 9997), incubated with indicated primary antibodies (see list below) diluted in 5% (wt/vol) of nonfat dry milk in 1X TBS-T, washed in 1X TBS-T, and incubated with appropriate secondary antibodies labeled with horseradish peroxidase (Jackson ImmunoResearch, Cat. 111035003 or 115035003) diluted in 5% (wt/vol) of nonfat dry milk in 1X

TBS-T. Protein bands were detected by chemiluminescence (Thermo Scientific, Cat. 32209, 34075, or 34095). Images were acquired using the C-Digit Blot scanner (LI-COR) and visualized in the Image Studio software (LI-COR). The following primary antibodies were used: anti-Arp2 (Abcam, Cat. ab128934); anti-Arp3 (Abcam, Cat. ab151729); anti-ARPC1A (Sigma-Aldrich, Cat. HPA004334); anti-ARPC1B (Sigma-Aldrich, Cat. HPA004832); anti-ARPC2 (Sigma-Aldrich, Cat. HPA008352); anti-ARPC3 (Sigma-Aldrich, Cat. HPA006550); anti-ARPC4 (Sigma-Aldrich, Cat. SAB2106287); anti-ARPC5 (Synaptic Systems, Cat. 305011); anti-ARPC5L (Abcam, Cat. ab169763); anti-phospho-STAT3 (Santa Cruz, Cat. sc-8059); anti-STAT3 (Santa Cruz, Cat. sc-482); anti-phospho-Zap-70 (Tyr319)/Syk (Tyr352) (Cell Signaling Technology, Cat. 2717); anti-Zap-70 (Cell Signaling Technology, Cat. 3165); anti-phospho-LAT (Tyr220) (Cell Signaling Technology, Cat. 3584); anti-phospho-PLCγ1 (Tyr783) (Cell Signaling Technology, Cat. 14008); anti- PLCγ1 (Cell Signaling Technology, Cat. 2822); anti-phospho-p44/42 MAPK (Erk1/2) (Thr202/Tyr204) (Cell Signaling Technology, Cat. 4377); anti-p44/42 MAPK (Erk1/2) (Cell Signaling Technology, Cat. 4695); anti-phospho-NF-κB p65 (Ser536) (Cell Signaling Technology, Cat. 3033); anti-NF-κB p65 (Cell Signaling Technology, Cat. 8242); anti-β-Actin (Cell Signaling Technology, Cat. 4970); anti- β-tubulin (Cell Signaling Technology, Cat. 2128); anti-GAPDH (Cell Signaling Technology, Cat. 2118); anti-Vinculin (Santa Cruz, Cat. sc-73614).

## Fluorescence microscopy of fibroblasts

P1's or HC's fibroblasts were seeded in 6-well plates pre-coated with RetroNectin (Takara, Cat. T100B) in complete DMEM medium and allowed to spread in a humidified incubator at 37 °C with 5% $CO_2$. After specified times, cells were washed in phosphate-buffered saline (PBS), fixed in 4% paraformaldehyde for 15 min, washed again, then permeabilized in 0.1% Triton X-100 in PBS for 10 min and blocked (10% FBS and 0.1% Triton X-100 in PBS) for 30 min. Cells were then incubated in blocking buffer with either Alexa Fluor 488 Phalloidin (Invitrogen, Cat. A12379), mouse anti-HA (Cell Signaling technology, Cat. 2367), or rabbit anti-Flag (Cell Signaling Technology, Cat. 14793 S) antibodies. Next, cells were washed in PBS. Samples stained with anti-HA and anti-Flag antibodies were incubated for 1 h with Alexa Fluor 594 or Alexa Fluor 568-conjugated secondary antibodies in blocking buffer (Invitrogen, Cat. A11072 and A21069). After washes, all samples were incubated with DAPI (Cell Signaling Technology, Cat. 4083) in PBS, for 10 min. Samples were washed twice in PBS and ready for imaging. Filopodia formation was best visualized 20 min after cell seeding and lamellipodia formation 120 min after cell seeding; therefore, those time points were selected for imaging. Images were collected using a ZOE fluorescent cell imager (Bio-Rad).

## Real-time cell analysis of adhesion and spreading

The rate and magnitude of adherence of fibroblasts was measured with the xCELLigence RTCA MP (Agilent, Santa Clara, CA). Briefly, E-plates 96 (Agilent, Santa Clara, CA) were background normalized after equilibrating the wells with 100 μL of complete media for 30 min. Following this step, $1 \times 10^4$ HC's or P1's fibroblasts were seeded and immediately placed onto the xCELLigence RTCA MP machine at 37 °C with 5% $CO_2$ to begin recording impedance every 1–2 min for 5 h and every 15 min thereafter. Impedance values were converted into the cell index by the RTCA software (Agilent, Santa Clara, CA) and plotted over time using the GraphPad Prism software version 8.3.0 (GraphPad, LLC). To measure spreading of neutrophils, isolated polymorphonuclear neutrophils ($1 \times 10^6$ cells/ml in 20 μL HBSS with $Ca^{2+}$ and $Mg^{2+}$ buffer (Gibco, Cat. 14025134) were added on the center of microscope slide. A coverslip was placed over the cells and digital images were captured at five second intervals for 600 s. The area and perimeter of the cells were measured using Infinity Analyze software (Lumenera, version 5.0.3).

## Cell migration assays

Fibroblasts migration was assessed with a wound healing (scratch) assay. Fibroblasts were seeded in a 12-well plate and serum-starved overnight. The next day, a scratch in the cell monolayer was generated with a pipette tip (20 μl tip, tip width 0.94 mm, Thermo Scientific Art tips). Cells were washed 2x in PBS and from then on cultured in DMEM supplemented with 2% FBS and 2 mM L-glutamine. Cells were imaged right after scratching (t = 0), at 8 h, and 24 h after the scratch was made. Images were captured with a Lionheart FX Automated Live Cell Imager (Agilent BioTeK) using a 4× objective with the high contrast brightfield accessory. To improve visualization, digital phase contrast processing was applied to entire images using Gen5 Image+ software (Agilent BioTeK). Neutrophil chemotaxis was monitored across a 260 μm platform separating the "cell" well from the "chemoattractant" well using the EZ-TAXIScan (Effector Cell Institute, Tokyo, Japan). Isolated neutrophils ($5 \times 10^3$ cells in 1.0 μl) were added to the "cell" well of the EZ-TAXIScan and 1.0 μl of either buffer (0.1% BSA in RPMI, 20 mM HEPES) or $5 \times 10^{-8}$ M fMLF (N-formylmethionyl-leucyl-phenyla-lanine, Sigma-Aldrich, Cat. F3506) was added to the opposing "chemoattractant" well. Digital images of the migrating PMNs were captured every 30 s for 1 h. Images were converted to stacks using the ImageJ software (version 1.53t; NIH). Ten randomly selected cells were electronically traced using the ImageJ plug-in, MTrackJ and the sequential positional coordinates of individual migrating cells were determined as a function of time. The tracks of individual migrating cells were reconstructed and plotted with the position of each cell anchored at the origin at t = 0. Since data were collected with time and position, multiple parameters could be derived – overall distance, directed distance (parallel to the chemoattractant, random distance (orthogonal to the chemoattractant), overall velocity, directed and random velocity vectors, and time-to-event analysis (number of cells completing migration and elapsed time). To assess migration in Jurkat cells, $5 \times 10^5$ wild-type (WT) or ARPC5-KO cells were suspended in 100 μL of RPMI 1640 media supplemented with 10% FBS (RPMI complete). The cells were loaded into 5.0 μm Transwell inserts (Corning) which were placed onto 24-well plates containing 450 μL of RPMI complete with or without 800 ng/mL CXCL12 (Peprotech). After 3 h of incubation at 37 °C, cells that had migrated to the lower chamber were collected and counted using an automatic cell counter (Countess II FL, ThermoFisher Scientific). Results were calculated as percentage of cells that had migrated out of total input cells.

## Rescue of wild-type ARPC5 expression

WT ARPC5 expression was rescued in P1's fibroblasts via plasmid transfection or lentiviral transduction. P1's fibroblasts were transfected with pCMV3-ARPC5-C-HA plasmid (Sino Biological, Cat. HG16494-CY) or mock-transfected (transient; transfection efficiency 20–40%) with empty vector using Amaxa Human Dermal Fibroblast Nucleofector Kit (Lonza, Cat. VPD-1001), program U-023, following manufacturer's protocol. For lentiviral transduction, $1.5 \times 10^5$ fibroblasts were seeded in a 6-well plate and cultured in a humidified incubator at 37 °C with 5% $CO_2$ overnight. Then, P1's cells were infected with pLenti-ARPC5-C-Myc-DDK-P2A-Puro viral particles (Origene, Cat. RC212631L3V) or control particles of pLenti-C-Myc-DDK-P2A-Puro (OriGene, Cat. PS100092V) at a multiplicity of infection (MOI) of 10, in the presence of polybrene (8 μg/mL). HC cells were infected with control particles following the same procedure. Next, cells were centrifuged at 1500 × g for 1 h. Media was exchanged the following day.

## Flow cytometry

For identification of surface proteins, single-cell suspensions of washed PBMC were stained with fluorochrome-conjugated antibodies (see list below) for 30 min at 4 °C. Then, cells were washed with stain buffer (BD Biosciences, Cat. 554656) twice and were ready for acquisition. For detection of the intracellular proteins FOXP3 and T-bet, cells were fixed

and permeabilized before staining, using the FOXP3/transcription factor staining buffer set (Invitrogen, Cat. 00552300), following manufacturer's recommendations. For STAT phosphorylation assays, PBMC with or without cytokine stimulation were surface stained with APC anti-CD4 antibody (BD Biosciences, Cat. 555349). Stimulated samples were incubated at 37 °C for 20 min with either IL-2 (10 ng/mL, Peprotech, Cat. 20002), IL-4 (10 ng/mL, Peprotech, Cat. 20004), IL-6 (10 ng/mL, Peprotech, Cat. 20006), IL-7 (10 ng/mL, Peprotech, Cat. 20007), IL-15 (100 ng/mL, Peprotech, Cat. 20015), IL-21 (50 ng/mL, Peprotech, Cat. 20021), IFNα (10 ng/mL, Cell Signaling Technology, Cat. 8927), IFNγ (10 ng/mL, Peprotech, Cat. 30002), or IL-6/IL-6R alpha protein chimera (Hyper IL-6, 100 ng/mL, R&D Systems, Cat. 8954SR). Next, cells were washed, fixed with BD Cytofix fixation buffer (BD Biosciences, Cat. BD554655), washed again, then permeabilized with BD Phosflow Perm Buffer III (BD Biosciences, Cat.558050). After washing, cells were stained with phospho-specific antibodies (see list below) for 1 h on ice. After two additional washes, cells were ready for acquisition. For STAT phosphorylation assays, monocytes were gated by FSC/SSC. All samples were acquired with BD FACSCanto II flow cytometer (BD Biosciences) using BD FACSDiva software. Further analyses of flow cytometry data were conducted in FlowJo v10.7.1 software (BD Biosciences). The following fluorochrome-conjugated antibodies were used: PE anti-IL-6Rα (Biolegend, Cat. 352804); APC anti-gp130 (Biolegend, Cat. 362006); APC-eFluor 780 anti-CD45 (Invitrogen, Cat. 47045942); APC-H7 anti-CD3 (BD Biosciences, Cat. 560176); BV510 anti-CD3 (BD Biosciences, Cat. 564713); Pacific Blue anti-CD3 (Biolegend, Cat. 344824); FITC anti-CD3 (Invitrogen, Cat. MHCD03014); PE-Cy7 anti-CD3 (BD Biosciences. Cat. 341091); APC anti-CD4 antibody (BD Biosciences, Cat. 555349); BV421 anti-CD4 (BD Biosciences, Cat. 565997); PE-Cy7 anti-CD4 (Biolegend, Cat. 300512); PerCP anti-CD4 (Invitrogen, Cat. MHCD0431); PerCP anti-CD4 (BD Biosciences, Cat. 566924); APC anti-CD8 (Invitrogen, Cat. MHCD0805); APC-R700 anti-CD8 (BD Biosciences, Cat. 565165); FITC anti-CD8 (BD Biosciences, Cat. 555366); FITC anti-CD45RA (Beckman Coulter, Cat. IM0584U); APC anti-CD62L (BD Biosciences, Cat. 559772); PE anti-CD25 (BD Biosciences, Cat. 555432); PE anti-CD25 (BD Biosciences, Cat. 341009); APC anti-FOXP3 (Invitrogen, Cat. 17477742); APC anti-CD19 (Invitrogen, Cat. 17019842); APC anti-CD19 (BD Biosciences, Cat. 555415); BB515 anti-CD19 (BD Biosciences, Cat. 564456); BV510 anti-CD19 (BD Biosciences, Cat. 562947); BV605 anti-CD20 (Biolegend, Cat. 302334); PE anti-CD21 (BD Biosciences, Cat. 557327); PE-Cy7 anti-CD21 (BD Biosciences, Cat. 561374); PE anti-CD24 (Invitrogen, Cat. 12024742); APC-R700 CD38 (BD Biosciences, Cat. 659118); BV421 anti-CD27 (BD Biosciences, Cat. 562513); FITC anti-Kappa (BD Biosciences, Cat. 643774); PE anti-Lambda (BD Biosciences, Cat. 642924); PerCP/Cy5.5 anti-IgM (Biolegend, Cat. 314512); PE-Cy7 anti-CD11c (Invitrogen, Cat. 25011642); PerCP/Cy5.5 anti-T-bet (Biolegend, Cat. 644805); FITC anti-CD14 (Invitrogen, Cat. MHCD1401); FITC anti-CD3/ PE anti-CD16+ CD56 (BD Biosciences, Cat. 340042); PE anti-Flag (Biolegend, Cat. 637310); Alexa Fluor 488 anti-STAT1 (pY701) (BD Biosciences, Cat. 612596); Pacific Blue anti-STAT3 (pY705) (BD Biosciences, Cat. 560312); Pacific Blue anti-STAT5 (pY694) (BD Biosciences, Cat. 560311); Alexa Fluor 488 anti-STAT6 (pY641) (BD Biosciences, Cat. 612600).

## Inhibitors
The Arp2/3 complex inhibitor I (CK-666, CAS 442633-00-3) was purchased from Milipore Sigma (Cat. 182515). Anti-IL6R antibody (rhPM-1 [tocilizumab]) was purchased from Novus Biologicals (Cat. NBP2-75192). Recombinant Human gp130 Fc Chimera Protein was purchased from R&D Systems (Cat. 671-GP).

## Quantification of IL-6 and IL-6Rα proteins
Whole blood samples collected into sodium heparin tubes were centrifuged for 15 min at 2000 × *g* for plasma separation. Plasma levels of soluble IL-6Rα were measured with Human premixed multi-analyte kit (R&D Systems, Cat. LXSAHM03) using the

Luminex 100/200 system (Luminex). Plasma levels of soluble IL-6Rα complexed with IL-6 were measured with Human IL-6/IL-6R alpha complex DuoSet ELISA (R&D Systems, Cat. DY8139-05) following manufacturer's instructions.

For detection of soluble IL-6Rα in CK-666 treated samples, CD4 cells were enriched by EasySep Human CD4 cell enrichment kit (Stemcell Technology, 19052). The enriched CD4 T cells (2 × 10^5 cells/ 100 μL) were either treated with vehicle control (DMSO) or CK-666 (100 μM) for 22 h. The supernatants were collected and soluble IL-6Rα was assessed by Luminex assay (R&D).

## Reporting summary
Further information on research design is available in the Nature Portfolio Reporting Summary linked to this article.

## Data availability
Whole exome sequencing datasets generated and analyzed during the current study are available in NCBI's Sequence Read Archive with accession no. PRJNA889418. Single-cell RNA sequencing raw and processed data have been deposited in NCBI's GEO data repository with accession no. GSE215451. T-cell receptor β sequencing data have been deposited in the ImmuneACCESS database [https://doi.org/10.21417/CJNS2023NC]. Source data are provided with this paper.

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

## Acknowledgements
This work was supported by the Intramural Research Program, National Institutes of Health Clinical Center and National Institute of Allergy and Infectious Diseases (to S.D.R.) and in part by federal funds from the National Cancer Institute, NIH, under Contract No. 75N91019D00024 (to D.B.K.). The content of this publication does not necessarily reflect the views or policies of the Department of Health and Human Services, nor does mention of trade names, commercial products, or organizations imply endorsement by the U.S. Government. The Laboratory of Human Genetics of Infectious Diseases is supported by the Howard Hughes Medical Institute, the Rockefeller University, the St. Giles Foundation, the National Institutes of Health (NIH) (R01 AI127564-06 to J.L.C. and A.P., R21AI159728 to B.Boi. & P01AI061093 to J.L.C.), the National Center for Advancing Translational Sciences (NCATS), NIH Clinical and Translational Science Award (CTSA) program (UL1TR001866), the French National Research Agency (ANR) under the "Investments for the Future" program (ANR-10-IAHU-01), the Integrative Biology of Emerging Infectious Diseases Laboratory of Excellence (ANR-10-LABX-62-IBEID to J.L.C.), the French Foundation for Medical Research (FRM) (EQU201903007798 to J.L.C.), the Square Foundation, Grandir - Fonds de solidarité pour l'enfance, Institut National de la Santé et de la Recherche Médicale (INSERM) and the University of Paris Cité. We acknowledge Drs. Andrew Issekutz and Dimas Mateos-Corral for their involvement in the care of P1.

## Author contributions
C.J.N.S., H.S.K., B.Boa., S.H., D.B.K., J.S., J.E.N., D.L.F., J.S.D., V.A.B., T.K., O.M.D., and M.B. designed and performed experiments. L.E.S., T.A.F., A.P., L.D.N., B.Boi., J.L.C., B.D., and S.D.R. supervised experiments. S.P. reviewed pathology slides. M.A.A. analyzed electron microscopy data. M.Ga. obtained informed consent from study participants. M.C.S., R.S., M.Gh., N.P., and B.D., provided patient samples and clinical information. C.J.N.S. wrote the first draft of the manuscript. All authors contributed to the final version of the manuscript. C.J.N.S. and S.D.R. planned and supervised the study.

## Funding

## Competing interests
The authors declare no competing interests.
