## [Peer Review File · Nature Communications]

Inherited ARPC5 mutations cause an actinopathy impairing cell motility and disrupting cytokine signalingREVIEWER COMMENTS

Reviewer #1 (Remarks to the Author):

Overall, I think this is a well-drafted and fascinating study. The study has two prominent novelties. First, the authors identify mutations in an important, yet underappreciated component of the ARP2/3 complex, ARPC5, that has not been previously linked to disease. They adequately showed that rescuing expression of ARPC5 reversed the observed phenotypes. The authors also linked the deficiency of ARPC5 to a deficiency of IL-6R α present on the cell surface, leading to poor cellular response to IL-6. Although I like the ideas and find their experimental data informative, I think more experiments can be done to further support and validate their current findings. Below are my detailed comments.

Major comments

The authors mention that P1's B cell populations were increased yet proliferated normally in stimulating conditions (page 6 lines 14 and 21). Given the autoantibodies found and the fact that the patient was on anti-CD20 treatment, the B cell compartment should be further investigated in my opinion. Leung et al. JCI Insight 2021 studied B cells of ARPC1B deficient patients. They observed many similar phenotypes such as failure to form lamellipodia and failure to spread. In addition, Leung et al. demonstrated tonic activation of the BCR and elevated phosphorylated ERK levels. I recommend the authors refer to the paper and see if similar phenomena are present in the ARPC5 patients given their lack of ARPC1B expression. Specifically, the authors can look at calcium signaling and ERK activation in the absence of ARPC5.

The authors identify that the lack of IL-6R α is the reason why ARPC5 deficient cells fail to respond to IL-6 in different cell types. The authors also attempted to demonstrate this with pharmacological inhibition in healthy cells. The concentration of CK-666 used is far above the IC₅₀ (Figure 4D). I would like for the authors to attempt lower, more relevant concentrations or at least address why the concentration and time point was used. A dose response curve would also be informative here.

In addition, the authors fail to explain the decrease in IL-6R α in response to ARP2/3 inhibition. Upon inhibiting ADAM17 with marimastat it was found that the decrease is not caused by the cleavage and secretion of IL-6R α (Figure 4D). It would be very useful to know what happens to the receptor after the two hours of inhibition (given that the decrease is not a result of using very high concentrations of the inhibitor). Alternative splicing using different RT-PCR probes can be investigated. The degradation of the protein using translational, or protease inhibitors can also be evaluated. Also, the level of soluble IL-6R α in the media can be measured after ARP2/3 inhibition using the same ELISA in Figure 4E.

Minor comments

Overall, figures were difficult to follow as they were not divided enough. For example, figure 2B has 6 panels, looking at two different cell types. In the text, (page 8 lines 6-13) Figure 2B is referred to 4 times and each time it references a different panel within the subfigure. Further dividing into more letters will help the reader better follow.

Error bars are difficult to see without zooming in on Figure 2B (bottom left), and 3B. Figure 2B (bottom right) is a better style that is easier to see and improves accessibility. Figure 2B (bottom right) is also missing error bars for the orange data. Number of replicates can also be added to the figure legend for improved transparency.

Scale bar is missing on Figure 2B top and middle right, as well as Figure 2C and 3C.

The figures also lack statistics, in Figure 4D for example.

Figure 1D is referred to on Page 5 lines 2 and 19, despite there being no mention of ARPC5 yet. Therefore, the structure and complex is irrelevant to the text.

Reviewer #2 (Remarks to the Author):

Nunes-Santos et al describe in this manuscript a novel autosomal recessive inborn error of immunity due to mutations in ARPC5 in two unrelated patients. They convincingly show that the patients phenotype is causally linked to the deficiency in ARPC5 by extensively studying one of the patients by a variety of molecular, biochemical, functional, imaging and phenotyping techniques, including genetic correction in patients' fibroblast. They also suggest a novel role of IL-6 signaling which could be linked to some of the disease manifestations.

Although these are only two cases and only one patient is studied biologically, the authors are to be commended for the amount and quality of data provided in support of the pathogenic role of the ARPC5 mutations.

I have a few points requiring clarification and discussion.

- 1) The conclusion on penetrance "Genetically, ARPC5 deficiency is inherited in an autosomal recessive manner with full clinical penetrance" should be down-toned since only two cases are described here.
- 2) The authors should discuss the possibility that the neurological phenotype is related to the underlying genetic disease or to a previous infection. Arp2/3 complex is well known to be involved in filopodia formation and neuritogenesis in neuronal cells (DOI: 10.1091/mbc.E21-06-0326). Is ARPC5 expressed in the brain?
- 3) Are there alternative explanations linked to innate immune function impairment for delay in wound healing in addition to fibroblast impaired migration ?
- 4) The authors should better discuss the platelet phenotype, which is most likely not due an intrinsic defect but due to autoimmunity.

Reviewer #3 (Remarks to the Author):

In their manuscript "Inherited ARPC5 deficiency is an actinopathy impairing cell motility and dissecting cytokine signaling" Nunes-Santos et al. describe the effect of germline biallelic null mutations in the ARPC5 subunit isoform of the Arp2/3 complex in two unrelated individuals. For one of the patients, availability of biospecimens allows describing how the loss of ArpC5 impacts overall health, immune reaction, cellular morphodynamics, protein expression levels and cytokine signaling.

The observation of ArpC5-deficiency as actinopathy in human patients is very interesting and the authors have performed a multitude of experiments to characterize the molecular effects of ArpC5 loss in different cell types. This, in principle, warrants publication. However, there are several points concerning the analysis on protein and cell biology level, which need to be addressed first.

Specifically, the authors show that in cells of patient 1 (P1), the absence of ARPC5 leads to reduced ArpC1A and ArpC1B levels and reduced amounts of complete Arp2/3 complexes. This correlates with the observed defects in cell morphology and cell migration and could thus also explain the observed immunological and developmental abnormalities found in the patients.

However, the characterization of protein levels shown in Western Blots requires quantitative analysis, in order to substantiate the conclusions. We acknowledge that quantitation using patient-derived primary cells might prove impossible. However, given that the authors have obtained an immortalized fibroblast cell line from P1, quantitative experiments using this cell line are needed to provide the required support to the presented data.

There is recent data available describing the effects of ARPC5 knock-downs (Abella et al., Ref. 20) and knock-outs (Faessler et al., 2023, DOI: 10.1126/sciadv.add6495) on cellular level. Those should be mentioned in the introduction as they highlight isoform-specific differences between ARPC5 and ARPC5L, which might also explain parts of the effects reported in the results section.

The analysis presented in Figure 2a,b is methodologically similar to what has been done by Faessler et al., 2023 (DOI: 10.1126/sciadv.add6495) for ARPC5, ARPC5L, and ARPC5/ARPC5L knockout

cells in rat fibroblasts and mouse melanoma cells. Interestingly, the provided phalloidin staining of P1's fibroblasts phenotypically appear to be more comparable to ARPC5/ARPC5L double knockout cells than ARPC5 single knockout cells (for example, when comparing the size of the fingerlike protrusions and the overall actin distribution). Also, in contrast to ARPC5 knockout B16-F1 mouse melanoma cells and Rat2 fibroblasts, P1's fibroblasts appear to not be upregulating ARPC5L levels to counteract the loss of ARPC5. This poses the question if the observed effects in the present study are indeed isoform specific or are related to overall reduced ARPC5/5L-levels. To clarify this matter, we suggest to expand the rescue experiments shown in Figure 3 to also include overexpression of ARPC5L and perform appropriate quantifications of the blots. This context should also be elaborated in the discussion.

ARPC2 expression levels appear not to be drastically altered in P1 in comparison to HCs (see Figure 2A, left panel). However, after ARPC5 expression in P1 fibroblasts, ARPC2 levels found in intact ARP2/3 complexes are much lower than in HC (Figure 3A). Since the bands associated with a higher electrophoretic mobility observed in the P1 empty vector control are absent in ARPC5 transfected cells, we are curious how the authors interpret this effect. Could loss of ArpC5 cause the existence of different complexes (such as one lacking ArpC5 and another lacking ArpC5 and ArpC1)? Would this explain the changed mobility and observed smear in the native blot? Quantification of results would be useful here.

As a main conclusion, the authors correlate the loss of ArpC5 to changed IL6 signaling and argue that the dysregulation of several, but not all Arp2/3-dependent functions is responsible for the cytokine-specific effect. Can the authors provide a better explanation, which Arp2/3-dependent functions they believe to be directly related to IL6-signalling and not the other unaffected pathways they looked at.

Also, we encourage the authors to compare their findings to a recently posted BioRxiv preprint describing ArpC5 deficiency in a patient and in vitro models (<https://doi.org/10.1101/2023.01.19.52468>).

Additional points:

Given that the employed ARPC5 antibody recognizes residues 15-19, could the authors provide more insights into how they made sure that this statement on page 7 is valid:

"N-terminus truncated proteins, resulting from alternative transcription initiation sites were not detected either."

The rescue expressions using transfection or lentiviral transduction use differently tagged ArpC5 constructs (HA and Myc-DDK tags). In Figure S9A, ArpC5 upon rescue in P1 runs quite high around 25kDa. Please specify in the figure and the figure legend which tag the detected protein contains. Is there another tag present, which is not described in the methods?

Figures 2B and 3B: One would expect that untreated P1's fibroblasts are comparable to the ones transfected with an empty vector. However, the presented phenotypes are vastly different. Are both representative images? If so, one would have to assume that the employed transfection protocol has a confounding impact on the performed experiments. Further, the standard deviations mentioned in the figure legend are not included in the actual figure.

The authors state on page 8:

"Treatment of HC's fibroblasts with the ARP2/3 inhibitor CK-666 resulted in reduced spreading capacity mimicking P1's fibroblasts behavior (Fig.S7)."

This is true in principle. Nevertheless, the spreading behavior of the untreated HC fibroblasts judged by the cell index is quite different to what is shown in Figure 2B. It should also be noted that when looking at the fibroblast phenotype observed for P1 (Figure 2B), it is dramatically different from the CK-666 treated fibroblasts of a HC (Figure S7A).

On page 9 the authors state:

“ARPC5 staining localized to the nucleus, to scattered dots throughout the cytoplasm, and to the edge of the plasma membrane on rescued cells, compatible with expected ARP2/3 complex subcellular localization (Fig.3b, Fig.S9).” ARPC5 isoform-specific subcellular localization has been described and this statement could therefore be backed up with citations.

Millard et al, 2003 (DOI: 10.1002/cm.10087)

Faessler et al., 2023 (DOI: 10.1126/sciadv.add6495)

The authors state on page 11:

“In contrast, ARPC1B deficiency does not seem to affect ARPC5 expression levels, suggesting that ARPC5 deficiency might have a broader impact on biology and disease.”

Could the authors explain their argument further? Is this because ARPC5 might also affect ARPC1A containing complexes? Because then ARPC1B might in return affect ARPC5L containing complexes.

The statement on page 12: “Branched actin is the predominant form of actin organization in lamellipodia, podosomes and invadopodia, all required for cell motility²¹.” This is not completely true as cell motility is possible without branched actin networks, for example in B16-F1 cells devoid of WAVE1/WAVE2, which do not exhibit lamellipodia (Tang et al., 2020, DOI: 10.1091/mbc.E19-12-0705). The same is true for leukocytes lacking one of the WRC subunits Hem1 (Leithner, Eichner et al., 2016 <https://doi.org/10.1038/ncb3426>).

The authors state:

“Whether Arp2/3 complex integrity in these scenarios is maintained by alternative isoforms of ARPC1 and ARPC5 or by non-canonical conformations of the complex lacking these subunits²⁶ is yet to be established.” We have already shown that complete Arp2/3 complexes containing specifically ARPC5L are devoid of ARPC5 (Faessler et al., 2023, DOI: 10.1126/sciadv.add6495).

Figure S9B: Why has ARPC1b been chosen for this blot? This does not represent all ARP2/3 complexes due to ARPC1 isoform diversity and is inconsistent with the main figures.

Minor comments

- Page 8: “Wild-type ARPC5 expression restored abnormal findings”: do the authors maybe mean “rescue abnormal findings”
- Figure 1D: the illustration of the branch junction does look off, with respect to how the different subunits are arranged (see for example recent work by the Pollard and Nolen labs on how the Arp2/3 complex is arranged within the branch junction). Maybe consider exchanging this panel.
- Figure 2A: the labeling of the native blot is confusing with “healthy control” being written side-by-side. Please arrange “healthy control” to be in one column.
- Figure 2B (right panels) and C: Panels lack scale bars.
- Figure 3C: Scale bars are missing
- Figure S8: The scale indication is hard to interpret. Please provide clearly visible scale bars. The corresponding supplementary materials and methods section is ambiguous about how the displayed specimen were prepared: “Poly-L-lysine-coated glass coverslips or glow-discharged carbon-coated glass coverslips were prepared, and cells were added to the coverslips and fixed again in 2% glutaraldehyde for 15 min.” Which of these preparations is shown.
- There seems to be a type in the following sentence: “The coverslip was mounted on an SEM stub and coated with a thin layer of approximately 20-30 nm of a gold-palladium in a vacuum evaporator for conductivity under electron beam and then imaged with Hitachi S-4500 field emission scanning electron microscope (FESEM).
- Figure 3A and S9B: the authors might want to consider merging these figures.

Methods and Supplementary Methods

- State the amount of protein loaded onto PAGE gels.
- Where possible, provide software version numbers and citations.

- Please provide manufacturers for all products mentioned in the methods section.
- Please state which milk powder and which buffer were combined to produce the blocking buffer, which buffer was used for washing and which buffer was used for diluting antibodies for Western blotting.
- Please state the composition of the blocking buffer employed for immunocytochemistry in the Methods section. Is it the same one as described in the Supplementary Methods?
- Full (non-cropped) Western blots should be provided as a supplementary figure.

Kind Regards,
Florian Fäßler and Florian Schur

REVIEWER COMMENTS

Reviewer #1 (Remarks to the Author):

Overall, I think this is a well-drafted and fascinating study. The study has two prominent novelties. First, the authors identify mutations in an important, yet underappreciated component of the ARP2/3 complex, ARPC5, that has not been previously linked to disease. They adequately showed that rescuing expression of ARPC5 reversed the observed phenotypes. The authors also linked the deficiency of ARPC5 to a deficiency of IL-6R α present on the cell surface, leading to poor cellular response to IL-6. Although I like the ideas and find their experimental data informative, I think more experiments can be done to further support and validate their current findings. Below are my detailed comments.

Response: We thank the Reviewer for the positive comments.

Major comments

The authors mention that P1's B cell populations were increased yet proliferated normally in stimulating conditions (page 6 lines 14 and 21). Given the autoantibodies found and the fact that the patient was on anti-CD20 treatment, the B cell compartment should be further investigated in my opinion. Leung et al. JCI Insight 2021 studied B cells of ARPC1B deficient patients. They observed many similar phenotypes such as failure to form lamellipodia and failure to spread. In addition, Leung et al. demonstrated tonic activation of the BCR and elevated phosphorylated ERK levels. I recommend the authors refer to the paper and see if similar phenomena are present in the ARPC5 patients given their lack of ARPC1B expression. Specifically, the authors can look at calcium signaling and ERK activation in the absence of ARPC5.

Response: Following the Reviewer's comment we evaluated the possibility of re-exploring P1's B cell constitutive signaling similarly as described by Leung et al., JCI Insight 2021 . Unfortunately, we did not have any remaining pre-Rituximab treated PBMCs. However, we did have one vial collected between the first 2 Rituximab infusions where we enriched B cells and tested BCR-induced downstream signaling by flow or immunoblotting (see figure below). Unlike the elevated BCR signaling observed in ARPC1B deficient cells by Leung et al., ARPC5 deficient P1 cells showed almost complete absence of baseline and BCR-induced phosphorylated ERK (p-ERK) and PLC γ 2 (p-PLC γ 2), similar to HEM1 deficient cells (Salzer et al, 2020 doi: 10.1126/sciimmunol.abc3979). In addition, P1's dad, an ARPC5 heterozygous mutation carrier, demonstrated a diminished (i.e., intermediate between P1 and controls) but not absent BCR-signaling/p-ERK and p-PLC γ 2 accumulation. These data would suggest that opposite to ARPC1B deficient cells which upregulate ARPC1A likely to, and as suggested by the authors, compensate for certain but not all Arp2/3 functions, ARPC5 deficient cells which simultaneously show absent ARPC5 and markedly decreased ARPC1B and ARPC1A proteins (Figure 1a and 3a) are unlikely able to do so. While we believe our data is reliable, it represents an n=1 experiment, it was performed on recently Rituximab-treated B cells (uncertain functional effect), and in a different experimental condition than in Leung et al. JCI paper (i.e., except for their baseline ARPC1B accumulation immunoblot on primary cells, they used patient-derived EBV-B cells or Ramos cell lines engineered to abolish ARPC1B expression (ARPC1B -KO Ramos cells) for all their functional tests). Under these circumstances, we opted to share this information with the Reviewers/Editor in the Point-by-Point response, refer and cite the paper by Leung et al., that we unintentionally omitted in the original submission, but do not include the experimental results in the revised version.

Figure legend. (A) B cells were enriched by EasySep Human B cell enrichment kit (Stemcell Technology, 19054), and the B cell purity was over 95%. Equal numbers of enriched B cells from each individual tested were then stimulated with a-IgM (20 ug/ml) for 15 minutes, and the total cell lysates were subjected to immunoblotting. **(B)**Total PBMCs were stained with an APC-cy7-CD19 antibody and stimulated with a-IgM for 15 minutes. After fixing and permeabilizing the cells using BD cytofix/cytoperm buffer, the cells were stained with p-PLC γ 2 antibody for 1h. CD19 B cells were gated, and the phosphorylation of PLC γ 2 (BD 558499) was analyzed by flow assay. Data are expressed as mean \pm SD from three different healthy donor controls.

The authors identify that the lack of IL-6R α is the reason why ARPC5 deficient cells fail to respond to IL-6 in different cell types. The authors also attempted to demonstrate this with pharmacological inhibition in healthy cells. The concentration of CK-666 used is far above the IC50 (Figure 4D). I would like for the authors to attempt lower, more relevant concentrations or at least address why the concentration and time point was used. A dose response curve would also be informative here.

Response: Response: Nolen et al., (Nature 460, 1031-1034; 2009) originally characterized CK-666 and other Arp2/3 complex inhibitors on homo sapiens, bovine, and yeast models. The authors established CK-666 IC50=4uM for Arp2/3 complex inhibition in the homo sapiens model to form actin filament comet tails in SKOV3 cells (human ovarian cancer cells) after infection with Listeria monocytogenes (yeast and bovine IC50=5uM and 17uM, respectively although under different experimental designs). Interestingly, in the homo sapiens model figures depicted in their paper, the authors needed 40uM (10x the IC50 dose described) incubation for 1h to inhibit actin polymerization. When they increased the inhibitors' concentration up to 100uM for 24h, neither class of compound tested (including CK-666) showed irreversible or off-target morphological effects on cells including apoptosis. Moreover, CK-666, the most potent compound they tested, had no effect on the mitotic index of human cells at concentrations up to 80 uM, while inhibiting actin assembly around the bacteria model completely at 10uM. In comparison to Nolen et al., when Fakim et al., (Front Pharmacol. 2022;13:896994) more recently analyzed several novel and more potent Arp2/3 inhibitors and compared them to CK-666, they established CK-666 IC50=56.29 (14x above Nolen et al., paper) also in a homo sapiens model; however, the cell type analyzed in their manuscript was different (i.e., human breast cancer MCF1 cell line) as well as the experimental design, incubation time and read-out (i.e., cell cycling progression determined by EdU incorporation after 18h of Arp2/3 inhibitors' exposure at ranges from 0-1000uM).

As clearly shown by the above mentioned and several other recent publications (Animals. 2023 Jan 12;13(2):263, 1; Br J Pharmacol. 2022;179:125–140; Am J Physiol Lung Cell Mol Physiol 2022;322: L662–L672; Biochem.Cell Biol. 2022;100: 458–472; Br J Cancer. 2023 doi: 10.1038/s41416-022-02135-4;

Mol Biol Cell. 2018;29:1465-1475; Tissue and Cell. 2023; 81:102028), 1) the target species (e.g., homo sapiens vs. bovine vs. ovine vs. yeast vs. crabs among several models), 2) the type of cell lineage used within the same species (e.g., human fibroblasts vs. human ovarian cancer cells vs. human myeloid cancer cells vs. human dendritic cells, among others) and 3) the experimental design and read-out (e.g., F-actin assembly and spindle organization in meiotic dynamics vs. blocking of lamellipodial protrusions vs. blocking triradiated spicules generation vs. reduction in a-smooth muscle actin and collagen-1 expression vs. branched filament-dependent epithelial to mesenchymal cell transition vs. decreased phagosomal acidification and increased antigen cross-presentation, among different experiments), all these on top of the Arp2/3 concentration and incubation time (ranging from 10 to 10,000uM, from 5min to 48h incubation) will determine the most appropriate and specifically tailored settings for each experimental design.

Closer to the ARPC5 disease we were studying, when focused on studies evaluating human ARPC1B deficiency (the best comparison to ARPC5 deficiency), Leung et al., (JCI Insight. 2021 Dec 8;6(23)) tested different ARPC1B cell functions in B cells and T cells using CK-666 at 800uM for up to 1h, and Randzavola et al (JCI. 2019;129(12):5600–5614) used CK-666 at 90uM for 10-25min to evaluate actin dynamics/centrosome distance to the synapse in a CTL cytotoxicity model when evaluating ARPC1B deficient cells.

In addition to the previously published experience, and as suggested by the reviewer, we did a (limited) dose curve testing CK-666 at 0uM (vehicle, DMSO), 50uM (almost identical IC50 as in Fakim et al., and 12.5x the IC50 as in Nolen et al., both established in human cells but at different experimental conditions and read outs), and 100uM (IC50 25x as in Nolen, et al., 2x as in Fakim et al., and the dose used in our experiments) (figure below). While we did not intend to establish a formal IC50 with this test, we did find that CK-666 100uM was a more effective dose than 50uM for our experimental design and readout.

Altogether, the CK-666 dose used in the experiment depicted in the new Figure 4d (100uM for 22h) was both within the effective but non-toxic dose range of several other studies in human cells have shown using this same Arp2/3 inhibitor.

Figure legend.

CD4 cells were enriched by EasySep Human CD4 cell enrichment kit (Stemcell Technology, 19052), and the CD4 cell purity was over 95%. The enriched CD4 T cells were either treated with vehicle control (DMSO) or the indicated concentration of CK-666 (50 or 100 uM) for 22 hours. The surface IL-6Ra

expression was tested by flow analysis. (B) The supernatants were collected from the experiment (A), and soluble IL-6R α was assessed by Luminex assay (R&D). Data are expressed as mean \pm SD from four different healthy donor controls.

In addition, the authors fail to explain the decrease in IL-6R α in response to ARP2/3 inhibition. Upon inhibiting ADAM17 with marimastat it was found that the decrease is not caused by the cleavage and secretion of IL-6R α (Figure 4D). It would be very useful to know what happens to the receptor after the two hours of inhibition (given that the decrease is not a result of using very high concentrations of the inhibitor). Alternative splicing using different RT-PCR probes can be investigated. The degradation of the protein using translational, or protease inhibitors can also be evaluated. Also, the level of soluble IL-6R α in the media can be measure after ARP2/3 inhibition using the same ELISA in Figure 4E.

Response: Following the Reviewer's suggestion, we did repeat and further expanded the IL-6R α biological fate experiments for the revised version. While reviewing the previous experiments and the methodological procedure description, we noticed that the experiment in Figure 4d was not described in full detail. The data on the original Figure 4d was generated after overnight incubation with CK-666, after which cells were washed and re-incubated for 2h with +/- CK-666 +/- marimastat. The decrease in the surface IL-6R α expression was likely due to the effect of CK-666, while the effect of marimastat was probably minimal since the cells had already been exposed to CK-666 overnight. In order to more accurately address the IL-6R α fate and the reviewer's questions, we tested cell surface IL-6R α expression on CD4 T cells, alternative splicing, and soluble IL-6R α in culture supernatants after treatment with CK-666 (Figure below). In this new experiment we determined that the surface IL-6R α expression was decreased by around 20% (panel A below), the relative gene expression levels of both membrane-bound IL-6R α and soluble IL-6R α (sIL-6R α) were comparable between the vehicle-treated (DMSO) and CK-666 treated samples, which rules out the possibility of alternative splicing associated with CK-666 treatment (panel B), and more interestingly, CK-666 treatment increased the levels of soluble IL-6R α in the supernatants (panel C), suggesting that the decreased surface IL-6R α depicted in panel A is likely due to increased cleavage of membrane-bound sIL-6R α . The updated data and explanation of the impact of CK-666 inhibitor on IL-6R α are now included in the revised manuscript.

While we did several attempts to add marimastat to this experiment, the pre-incubation with CK-666 seemed to prevent any inhibitory effect of ADAM17, and therefore we excluded it from the revised version of the manuscript.

Figure legend.

CD4 cells were enriched by EasySep Human CD4 cell enrichment kit (Stemcell Technology, 19052), and CD4 cell purity was >95%. The enriched CD4 T cells were either treated with vehicle control (DMSO) or CK-666 (100 μ M) for 22 hours. The surface IL-6R α expression was tested by flow analysis. (B) Total RNA from the CD4 cells was extracted using the RNeasy Mini kit (Qiagen), and the cDNA was generated using reverse Transcriptase (Qiagen) according to the manufacturer's protocol. Quantitative RT-PCR was performed using SYBR Green Dye (Applied Biosystems) on the StepOne Plus (Applied Biosystems). Relative gene expression was calculated and normalized to the vehicle treated membrane-bound IL6R α (mIL6R α). (C) The supernatants were collected and soluble IL-6R α was assessed by Luminex assay (R&D). Data are expressed as mean \pm SD from five different healthy donor controls.

Minor comments

Overall, figures were difficult to follow as they were not divided enough. For example, figure 2B has 6 panels, looking at two different cell types. In the text, (page 8 lines 6-13) Figure 2B is referred to 4 times and each time it references a different panel within the subfigure. Further dividing into more letters will help the reader better follow.

Response: For clarity, Figures are now further divided and better identified to simplify the readers' experience.

Error bars are difficult to see without zooming in on Figure 2B (bottom left), and 3B. Figure 2B (bottom right) is a better style that is easier to see and improves accessibility. Figure 2B (bottom right) is also missing error bars for the orange data. Number of replicates can also be added to the figure legend for improved transparency.

Scale bar is missing on Figure 2B top and middle right, as well as Figure 2C and 3C.

Response: Error and scale bars, as well as the information on number of replicates per experiment have been more clearly added to the revised version.

The figures also lack statistics, in Figure 4D for example.

Response: In our manuscript, we show data from a single patient with technical replicates, which makes it difficult to determine significant differences (i.e., in the revised Figure 2c, 2d, 4e). Moreover, in Figure 4d, to account for variations between different experiments, we normalized the data to Vehicle/Untreated sample controls. Since the control group displayed no variability due to the normalization (SD=0), it is challenging to establish statistical significance in those samples. Therefore, we included individual values either in the graphs or the source data file.

Figure 1D is referred to on Page 5 lines 2 and 19, despite there being no mention of ARPC5 yet. Therefore, the structure and complex is irrelevant to the text.

Response: We included Figure 1D depicting the Arp2/3 complex and function as a context reference for the readers; as both R1 and R3 found it odd we decided to remove it from the revised version.

Reviewer #2 (Remarks to the Author):

Nunes-Santos et al describe in this manuscript a novel autosomal recessive inborn error of immunity due to mutations in ARPC5 in two unrelated patients. They convincingly show that the patients phenotype is causally linked to the deficiency in ARPC5 by extensively studying one of the patients by a variety of molecular, biochemical, functional, imaging and phenotyping techniques, including genetic correction in patients' fibroblast. They also suggest a novel role of IL-6 signaling which could be linked to some of the disease manifestations.

Although these are only two cases and only one patient is studied biologically, the authors are to be commended for the amount and quality of data provided in support of the pathogenic role of the ARPC5 mutations.

I have a few points requiring clarification and discussion.

Response: We sincerely thank the Reviewer for the positive comments.

1) The conclusion on penetrance "Genetically, ARPC5 deficiency is inherited in an autosomal recessive manner with full clinical penetrance" should be down-toned since only two cases are described here.

Response: Following the Reviewer's comment, the sentence was modified as follows "Genetically, ARPC5 deficiency is inherited in an autosomal recessive manner with no evidence of haploinsufficiency (i.e., heterozygous parents were asymptomatic); while clinical penetrance seems complete (i.e., both individuals with biallelic mutations were symptomatic), more patients will have to be described to fully confirm disease penetrance and expressivity."

2) The authors should discuss the possibility that the neurological phenotype is related to the underlying genetic disease or to a previous infection. Arp2/3 complex is well known to be involved in filopodia formation and neuritogenesis in neuronal cells (DOI: 10.1091/mbc.E21-06-0326). Is ARPC5 expressed in the brain?

Response: The Reviewer raises a very important point about CNS involvement in ARPC5 deficiency. ARPC5 is indeed ubiquitously transcribed and expressed in most human organs or systems, including the CNS (<https://www.genecards.org/cgi-bin/carddisp.pl?gene=ARPC5>). In our cohort, we report P1/Family1 with syndromic manifestations but no CNS involvement (actually, the best student in her class), and P2/Family2 with syndromic manifestations including severe CNS involvement. While we highly suspect P2's eldest sister/Family 2 was also affected with ARPC5 deficiency and did not present with syndromic features or CNS complications by the time of her demise (6 months), we have no confirmatory laboratory proof of her genetic status. We did find P2 was homozygous for a missense variant in *BRINP2* (NM_021165 c.184C>T, p.R62W; CADD score of 30; not found in gnomAD, heterozygous in her parents), and possibly associated with her neurodevelopmental disorders. Brinps 1-3, and Astrotactins (Astn) 1 and 2 are members of the Membrane Attack Complex/Perforin (MACPF) superfamily that are predominantly expressed in the mammalian brain during development. Genetic variation at the human *BRINP2/ASTN1* and *BRINP1/ASTN2* loci has been implicated in neurodevelopmental disorders. BMP/RA-inducible neural-specific protein(s) (Brinps) are a family of three highly conserved vertebrate genes that are almost exclusively expressed in neurons in the central and peripheral nervous system (Kawano et al., 2004; Terashima et al., 2010). *Brinp2* and *Brinp3* are expressed in differentiated neurons of the neocortex, amygdala, hippocampus, and cerebellum during mammalian brain development. Moreover, Terashima et al. (2010) showed that expression of all 3 *BRINP* proteins suppressed cell-cycle progression of mouse neural stem cells (NSCs). Based on the *BRINP2* genetic data detected in P2 and the previously reported data described, we based our statement on the possible association with her CNS manifestations. From a different perspective, we explored the prevalence of CNS manifestations in patients with *ARPC1B* deficiency, the closest disease model to *ARPC5* deficiency. *ARPC1B* deficiency has been reported in at least 34 patients, and CNS manifestations are not prevalent except when vascular events, mostly related to thrombocytopenia, are considered (Vazquez-Echeverri et al., *J Allergy Clin Immunol Pract.* 2023 Jan 25:S2213-2198(23)00085-5.). When focused on structural/developmental CNS complications, we only found 1 out-of-the 34 *ARPC1B* deficient patients reported (pt.6; Brigida et al., *Blood.* 2018;132(22):2362-2374) presenting with "syndromic features, as fine and brittle hair and keratotic hair lesions, a peculiar face, big thumb and hallux, dysplastic teeth, delay of dental eruption, and early release of the upper gums were evident since birth. Microcephaly (-2SD) and psychomotor retardation (i.e. independent walking at 3 years), delay of language acquisition and learning difficulties were observed". The parents, self-reported as non-consanguineous, both carried the extremely rare variant c.64+1G>A (gnomAD, 3/268588 alleles tested; allele frequency 0.00001) that was found in homozygous state on their affected daughter with severely decreased *ARPC1B* expression. Interestingly, the same rare variant was found by the same group on an unrelated patient (pt.1) in their cohort. While this other patient was also severely ill, no reports of syndromic features or structural/developmental CNS complications were noted. To add more information to the Reviewer's query, a recent preprint by Sindram et al., (<https://doi.org/10.1101/2023.01.19.524688> ; available online ~2 months following submission of our manuscript to the Nature group) describes 2 Iranian siblings, 1 confirmed (female) and 1 suspected (male) to carry novel homozygous null *ARPC5* variants. Both siblings presented with severe invasive infections and abscesses since birth and died early in life. Clinically, the patient confirmed to carry the biallelic *ARPC5* null variants was born with a congenital heart disease and did not have any CNS manifestations. Her sibling, suspected of carrying the same mutants, was also born with a congenital heart disease, plus a cleft palate, and CNS structural defects (i.e., hypoplastic corpus callosum and hydrocephalus). With no patient cells or biologic material available for further testing (preprint text "Unfortunately, more detailed additional immunophenotyping to investigate parameters such as lymphocyte subsets, functional responses to mitogens and antigens, serum cytokine levels, or markers of type I and II interferon and/or NF-kappaB

activation, was not possible due to the early deaths of both children”), the authors generated a CRISPR-Cas9 *Arpc5*^{-/-} null murine model to study the disease, that showed embryonic lethality at 9.5 dpf with incomplete cranial neuropore closure, heart and 2nd pharyngeal arch formation. Based on the clinical/laboratory data available in the preprint, the 2 siblings (assuming both were genetically affected by ARPC5 deficiency) do share several features with the patients in our study. In terms of the murine model generated, it shows disrupted tissue morphogenesis and an embryonic lethality, therefore preventing any immune studies to be performed. This contrasts with the human disease that demonstrates viable fetuses, although presenting with severe diseases since birth, and at least 1 patient surviving into her 2nd decade of life (P1, our manuscript). Of note, no structural cardiovascular diseases were detected in either ARPC5 mutated patients in our manuscript.

In summary, with the limited evidence available (i.e., number of patients reported so far) we can neither confirm nor rule out the CNS structural/neurodevelopmental manifestations relationship with ARPC5 deficiency in humans. However, it seems safe to state that syndromic manifestations are more likely to occur in ARPC5 deficiency than in ARPC1B deficient patients. This assertion could be substantiated by the Arp2/3 expression pattern (ubiquitous) but the broader impact of ARPC5 deficiency in the complex (also decreasing ARPC1A and ARPC1B levels) but not the other way around. In addition, if ARPC5 deficiency and CNS manifestations are indeed proven to be linked, the clinical penetrance of this complication is definitively not complete. A brief summary of this data is now included in the Discussion section of the revised MS.

3) Are there alternative explanations linked to innate immune function impairment for delay in wound healing in addition to fibroblast impaired migration ?

Response: The reviewer raised an important point regarding mono- or multi-layer causality for the wound healing defects in ARPC5 deficient patients. Wound healing is a natural physiological reaction to tissue injury. However, wound healing is not a simple phenomenon but rather involves a complex interplay between numerous cell types, cytokines, mediators, and the vascular system. The cascade of initial vasoconstriction of blood vessels and platelet aggregation is designed to stop bleeding. This is followed by an influx of a variety of inflammatory cells, starting with the neutrophil. These inflammatory cells, in turn, release a variety of mediators and cytokines to promote angiogenesis, thrombosis, and re-epithelialization. The fibroblasts, in turn, lay down extracellular components which will serve as scaffolding for wound healing (Ozgok Kangal MK, Regan JP. StatPearls [Internet]. StatPearls Publishing; Treasure Island (FL): May 8, 2022). In the context of the above “wound healing model”, fibroblast filling a scratch on a tissue culture dish is both a well-established, as well as an oversimplified single cell lineage method to determine abnormalities in the wound healing process. Focusing on PID/IEI with abnormal wound healing, chronic granulomatous disease (CGD) and leukocyte adhesion deficiency type 1 (LAD1) are frequently associated with this feature. CGD is caused by abnormal NADPH-oxidase respiratory burst leading to hyperinflammation and granuloma formation, that in its turn prevent appropriate wound healing; LAD1 patients have a CD18/ β 2-integrin deficiency preventing proper formation of CD11a/CD18, CD11b/CD18 and CD11c/CD18 heterodimers. This impairs neutrophil firm adhesion to endothelial cells and blood stream egress towards the wound, preventing their critical role in wound healing. In this scenario, it is certainly possible (and highly likely) that 1) the baseline systemic hyperinflammatory status and 2) the intrinsic neutrophil adhesion and migration defect observed in patients with ARPC5 deficiency

contribute, together with the fibroblast defects, to the abnormal wound healing process observed in these patients. A summarized version of this explanation is added to the revised version of the manuscript.

4) The authors should better discuss the platelet phenotype, which is most likely not due an intrinsic defect but due to autoimmunity.

Response: We appreciate the Reviewer's suggestion as it allow us to further discuss this manifestation. Thrombocytopenia in ARPC5 deficiency is likely multifactorial (i.e., intrinsic and extrinsic) with different lines of evidence directly or indirectly supporting this hypothesis. P1 had at least 3 bone marrow biopsies performed during her life in the context of peripheral blood thrombocytopenia. In those studies, her megakaryocytes were not found to be increased but twice were found to be decreased and showing dysplastic nuclear features with a low immature platelet fraction, suggesting that production was decreased. Although not absolute proof of platelet intrinsic involvement in ARPC5 deficiency, patients with ARPC1B deficiency (ARPC1B protein expression is markedly decreased in ARPC5 deficiency), show microthrombocytopenia with aberrant spreading; and knocking down ARPC1B in megakaryocytic cells resulted in decreased proplatelet formation (Nat Commun. 2017; 8: 14816). P1 in our manuscript also had hepatosplenomegaly since early in life that could have resulted in hypersplenism with platelet sequestration and consumption. Furthermore, although not formally detected, it would not be surprising that antiplatelet antibodies could have also been part of the broad autoantibody profile exhibit by P1 and could have contributed to her thrombocytopenia, as suggested by the reviewer. A brief summary of this data is now included in the Discussion of the revised version.

Reviewer #3 (Remarks to the Author):

In their manuscript "Inherited ARPC5 deficiency is an actinopathy impairing cell motility and dissecting cytokine signaling" Nunes-Santos et al. describe the effect of germline biallelic null mutations in the ARPC5 subunit isoform of the Arp2/3 complex in two unrelated individuals. For one of the patients, availability of biospecimens allows describing how the loss of ArpC5 impacts overall health, immune reaction, cellular morphodynamics, protein expression levels and cytokine signaling.

The observation of ArpC5-deficiency as actinopathy in human patients is very interesting and the authors have performed a multitude of experiments to characterize the molecular effects of ArpC5 loss in different cell types. This, in principle, warrants publication. However, there are several points concerning the analysis on protein and cell biology level, which need to be addressed first.

Response: We thank Dr. Florian Fäßler and Dr. Florian Schur, both reviewers involved in "Reviewer 3" for their detailed evaluation and comments.

Specifically, the authors show that in cells of patient 1 (P1), the absence of ARPC5 leads to reduced ArpC1A and ArpC1B levels and reduced amounts of complete Arp2/3 complexes. This correlates with the observed defects in cell morphology and cell migration and could thus also explain the observed immunological and developmental abnormalities found in the patients.

However, the characterization of protein levels shown in Western Blots requires quantitative analysis, in order to substantiate the conclusions. We acknowledge that quantitation using patient-derived primary

cells might prove impossible. However, given that the authors have obtained an immortalized fibroblast cell line from P1, quantitative experiments using this cell line are needed to provide the required support to the presented data.

Response: To address the Reviewers' comment, we added protein densitometry measures normalized to housekeeping proteins to accurately quantify the impact of the ARPC5 biallelic variants onto the specific Arp2/3 complex components. Densitometry measures overall correlated with our previous conclusions showing absent ARPC5 expression (0/10 of HC T cell blasts and fibroblasts), markedly reduced ARPC1A and ARPC1B expression (1-5 [for both proteins]/10 of HC T cell blasts and fibroblasts), and limited impact on ARPC5L expression (7-13/10 of HC T cell blasts and fibroblasts), to present the most relevant findings.

There is recent data available describing the effects of ARPC5 knock-downs (Abella et al., Ref. 20) and knock-outs (Faessler et al., 2023, DOI: 10.1126/sciadv.add6495) on cellular level. Those should be mentioned in the introduction as they highlight isoform-specific differences between ARPC5 and ARPC5L, which might also explain parts of the effects reported in the results section.

Response: As mentioned by the Reviewers, Ref. 20 is already included in our reference list, and the SciAdv paper published by the Reviewers has been added to the references in the revised manuscript. Of note, the paper by Faessler and Schur (SciAdv, 2023) was published on 01/20/2023, 1.5 months after we submitted our manuscript for evaluation on 12/06/2022. In fact, having known the content of the Reviewers' SciAdv paper before our submission would have certainly facilitated our work and enriched our discussion.

The analysis presented in Figure 2a,b is methodologically similar to what has been done by Faessler et al., 2023 (DOI: 10.1126/sciadv.add6495) for ARPC5, ARPC5L, and ARPC5/ARPC5L knockout cells in rat fibroblasts and mouse melanoma cells. Interestingly, the provided phalloidin staining of P1's fibroblasts phenotypically appear to be more comparable to ARPC5/ARPC5L double knockout cells than ARPC5 single knockout cells (for example, when comparing the size of the fingerlike-protrusions and the overall actin distribution). Also, in contrast to ARPC5 knockout B16-F1 mouse melanoma cells and Rat2 fibroblasts, P1's fibroblasts appear to not be upregulating ARPC5L levels to counteract the loss of ARPC5. This poses the question if the observed effects in the present study are indeed isoform specific or are related to overall reduced ARPC5/5L-levels. To clarify this matter, we suggest to expand the rescue experiments shown in Figure 3 to also include overexpression of ARPC5L and perform appropriate quantifications of the blots. This context should also be elaborated in the discussion.

Response: The model presented and studied in our manuscript is a human disease molecularly characterized by biallelic ARPC5 null mutations. While the Reviewers perceived close similarities in terms of actin distribution between the human P1 fibroblasts in our experiments and the Reviewers' models of Arpc5/Arpc5L double knockouts of rat fibroblasts and malignant (melanoma) murine cells, we have extensively showed that human fibroblasts, T cell blasts and PBMC carrying germline ARPC5 null mutations do not express ARPC5, have reduced ARPC1A and ARPC1B expression, and efficiently express ARPC5L at not reduced levels when compared to the healthy controls (Figures 2a, 3a, S6a, S6b, S9a). Moreover, when ARPC5 expression was rescued in human ARPC5-null fibroblasts, no changes in ARPC5L levels were detected, opposed to ARPC1A and ARPC1B levels that were indeed normalized upon correction (Figure 3a, S9a). Based on the reviewer's request, we also overexpressed the ARPC5L protein in P1 fibroblasts to investigate whether this had any effect on ARPC5, ARPC1A, or ARPC1B expression. In the experiment depicted below we demonstrate that ARPC5L overexpression

does not exert relevant effects on those proteins' expression: i.e., ARPC5, ARPC1A and ARPC1B levels were not altered after ARPC5L was efficiently transfected and overexpressed. Our analysis of human primary cells (P1 fibroblasts) overexpressing ARPC5L data strongly suggests that the primary defect in ARPC5 deficiency is associated with ARPC5 (plus ARPC1A and ARPC1B) reduced expression, rather than lack of ARPC5L expression.

Figure legend

The patient's fibroblasts were transfected with pCMV6-Myc-DDK-ARPC5L (Origene, RC201699) using Amaxa Nucleofector kit (Lonza, VPD-1001, program U-023). After 24 hours, cell lysates were prepared and tested for indicated protein expressions. Vinculin was used as a loading control. The numbers below the western blotting images (ARPC5, ARPC1A and 1B) represent the protein expression levels, quantitatively measured in relation to healthy controls, after normalization to the loading control. Empty vector (EV) transfected Normal controls value was set at 100.

ARPC2 expression levels appear not to be drastically altered in P1 in comparison to HCs (see Figure 2A, left panel). However, after ARPC5 expression in P1 fibroblasts, ARPC2 levels found in intact ARP2/3 complexes are much lower than in HC (Figure 3A). Since the bands associated with a higher electrophoretic mobility observed in the P1 empty vector control are absent in ARPC5 transfected cells, we are curious how the authors interpret this effect. Could loss of ArpC5 cause the existence of different complexes (such as one lacking ArpC5 and another lacking ArpC5 and ArpC1)? Would this explain the changed mobility and observed smear in the native blot? Quantification of results would be useful here.

Response: The Reviewers' observation is technically correct; however, these results need to be interpreted via a different context perspective (e.g., cell source contribution and rescue efficiency). Figure 2a shows Arp2/3 complex individual protein accumulation levels from lysates extracted from T cell blasts. The cell source is a major point in the data interpretation, as 100% of these cells contributed to the protein lysates meaningfully analyzed in this experiment. In comparison, in Figure 3a and b, we evaluated P1 fibroblasts transiently transfected or transduced with empty or WT ARPC5 vectors. While all cells in this experiment were exposed to the same conditions (e.g., transfection or transduction) and lysed, only those efficiently corrected would contribute to ARPC5 levels, and eventually to all the other components of those particular Arp2/3 complexes, specifically ARPC2.

In terms of the Arp2/3 complexes with higher mobility shift observed on the un-rescued P1 fibroblasts in Figure 3b (empty vector transduced), we fully agree with the Reviewers that they likely represent alternative/aberrant ARPC5-deprived Arp2/3 complexes. The occurrence of hybrid complexes comprised of some Arp2/3 complex subunits plus other proteins (e.g., vinculin) has been demonstrated by mass spectrometry in focal adhesions from chicken smooth muscle (Chorev et al. 2014 <https://doi.org/10.1038/ncomms4758>). These authors also observed that disturbance of individual Arp2/3 complex subunits in the same model (either by knockdown or overexpression) influenced the quantity and quality of the hybrid complexes. The finding that the aberrant/alternative Arp2/3 complexes were no longer detected when P1 fibroblasts were efficiently rescued (WT ARPC5 vector-transduced) and replaced by Arp2/3 complexes of identical mobility shift as the HC's, further corroborated this hypothesis. A summary of this interpretation is further explained in the revised manuscript.

As a main conclusion, the authors correlate the loss of ArpC5 to changed IL6 signaling and argue that the dysregulation of several, but not all Arp2/3-dependent functions is responsible for the cytokine-specific effect. Can the authors provide a better explanation, which Arp2/3-dependent functions they believe to be directly related to IL6-signalling and not the other unaffected pathways they looked at.

Response: When evaluating the IL-6 signaling pathway we demonstrated that the fate, and in its turn the function, of IL-6R α was severely affected, but not that of gp130 (its co-receptor in IL-6 signaling) (Figure 4a, b and c). We also demonstrated that signaling through the IL-2 receptor (IL2R α , IL-2R β and γ -C), the IL-7 receptor (IL-7R α , γ -C), the IL-15 receptor (IL-15R α , γ -C), IL-21 receptor (IL-21R α , γ -C), and the IL-4 receptor (IL-4R α , γ -C; IL-4R α , IL-13R α) on CD4 T cells, or the IL-4 receptor (IL-4R α , IL-13R α), the IFN α receptor (IFN α R1, IFN α R2) and the IFN- γ receptor (IFN γ R1, IFN γ R2) on monocytes, remained intact and unaffected (Figure S10). Therefore these results suggest a specific role for ARPC5 involvement in IL-6 signaling vs. the other cytokines evaluated. Along the same line, when we tested different processes known to require functional Arp2/3 complexes to maintain integrity (e.g., T cell proliferation (Figure S1), TCR mediated signaling (Figure S12), microcluster formation and activation-induced actin rearrangement (Figure S13), as well as T cell activation by antigen presenting cells), all those functions were preserved in the absence of ARPC5, strongly suggesting that this protein was redundant in the Arp2/3 complex for those particular functions, but not for others like neutrophil and fibroblast adhesion and migration (Figure 2b,c).

Also, we encourage the authors to compare their findings to a recently posted BioRxiv preprint describing ArpC5 deficiency in a patient and in vitro models (<https://doi.org/10.1101/2023.01.19.52468>).

Response: While we are open to discuss preprinted material in the Point-by-Point response to the Editor and the Reviewers, we are not in favor of validating and citing not yet peer-reviewed/unpublished scientific material (both features intrinsic to preprints) in the final version manuscript when describing a novel disease. In that context, the recent preprint by Sindram et al., (<https://doi.org/10.1101/2023.01.19.524688>; only available online ~2 months after our manuscript was submitted for evaluation at the Nature group) describe 2 Iranian siblings, 1 confirmed (female) and 1 suspected (male) to carry novel homozygous null ARPC5 variants. Both siblings presented with severe invasive infections and abscesses since birth and died early in life. Clinically, the patient confirmed to carry the biallelic ARPC5 null variants was born with a congenital heart disease and did not have any CNS manifestations. Her sibling, suspected of carrying the same mutants, was also born with a congenital heart disease, plus a cleft palate, and CNS structural defects (i.e., hypoplastic corpus callosum and hydrocephalus). With no patient cells or biologic material available for further immune testing (quote “Unfortunately, more detailed additional immunophenotyping to investigate parameters such as lymphocyte subsets, functional responses to mitogens and antigens, serum cytokine levels, or markers of type I and II interferon and/or NF-kappaB activation, was not possible due to the early deaths of both children”), the authors generated a CRISPR-Cas9 *Arpc5*^{-/-} null murine model to study the disease, that showed embryonic lethality at 9.5 dpf with incomplete cranial neuropore closure, heart and 2nd pharyngeal arch formation. Based on the clinical/laboratory data available in the preprint, the 2 siblings (assuming both were genetically affected by ARPC5 deficiency) do share several features with the patients in our study. In terms of the murine model generated, it shows disrupted tissue morphogenesis and an embryonically lethal phenotype, therefore preventing any immune studies to be performed. The embryonic lethality contrasts with the human disease that demonstrated viable fetuses, although presenting with severe diseases developing in infancy, as well as at least 1 patient surviving into her 2nd decade of life (P1, our manuscript). Of note, while no structural cardiovascular diseases were detected in the ARPC5 mutant patients in our manuscript, we do believe that syndromic features are likely part of the phenotype of this ARPC5 deficiency.

Additional points:

Given that the employed ARPC5 antibody recognizes residues 15-19, could the authors provide more insights into how they made sure that this statement on page 7 is valid:

“N-terminus truncated proteins, resulting from alternative transcription initiation sites were not detected either.”

Response: While in the WB figures in our manuscript we only showed ARPC5 detected by a monoclonal ab, we also tested a rabbit polyclonal ARPC5 Ab (Abcam, ab118459). This commercially available Ab was generated from a synthetic peptide corresponding to the first 100 amino acids of human ARPC5, and therefore allowing for detection of alternative isoforms of the protein. The ARPC5 monoclonal and polyclonal Abs, together with an ARPC5L mAb, were tested in parallel on cell lysates from HC and P1 fibroblasts, as well as on EV-, ARPC5L- or ARPC5-transfected HEK293 cells. As presented below in figure (a), the ARPC5 mAb was able to specifically detect ARPC5 in HC fibroblasts, but not in ARPC5-deficient P1 fibroblasts (no alternative -higher or lower MW- bands were evidenced either); the ARPC5 mAb also specifically detected endogenous ARPC5, as well as overexpressed ARPC5 in HEK293T transfected cells. In contrast to the ARPC5 mAb, the polyclonal ab in figure (b) detected ARPC5L in ARPC5L-transfected HEK293T cells as well a single and non-specific ~17kDa band in the ARPC5-deficient P1 fibroblasts (likely endogenous ARPC5L). When the ARPC5L mAb was tested in

figure (c), native ARPC5L was detected at similar levels in all cell lysates, and in ARPC5L-transfected HEK293T cells. Altogether these data strongly suggests that 1) no alternative ARPC5 bands were detected in the ARPC5-deficient fibroblasts from P1, either by a monoclonal or a polyclonal ARPC5 ab, and 2) the ARPC5 polyclonal ab by Abcam is sensitive to detect its target protein but cross-reacts with ARPC5L in its native and overexpressed forms. Since then, and likely due to its cross-reactivity, Abcam discontinued their ARPC5 rabbit polyclonal ab118459 product from the market.

Figure legend

HEK293T cells were transfected with the indicated pCMV6 Myc-DDK vector expressing either ARPC5L or ARPC5 (EV; empty vector). After 72 hours of incubation, cell lysates were prepared, and immunoblotting was performed with the indicated antibody. Fibroblasts from a healthy control and P1 were used as positive/negative controls and to show endogenous levels of the proteins.

The rescue expressions using transfection or lentiviral transduction use differently tagged ArpC5 constructs (HA and Myc-DDK tags). In Figure S9A, ArpC5 upon rescue in P1 runs quite high around 25kDa. Please specify in the figure and the figure legend which tag the detected protein contains. Is there another tag present, which is not described in the methods?

Response: The ARPC5 protein consists of 151 amino acids. By using the pCMV6-Myc-DDK vector, an additional 31 amino acids (including the cloning site, MluI) are also expressed, adding ~3.4 kDa to the

~17 kDa endogenous ARPC5 protein, and therefore resulting in a ~20.4 kDa protein. When we re-analyzed the originally submitted WB images, we noticed that the protein molecular weight labeling was slightly off, as it appeared to be close to 25 kDa, instead of the actual size of just over 20 kDa. We adjusted the protein marker labeling accordingly and made a note of the difference in protein size in the revised Figure S9 legend.

Figures 2B and 3B: One would expect that untreated P1's fibroblasts are comparable to the ones transfected with an empty vector. However, the presented phenotypes are vastly different. Are both representative images? If so, one would have to assume that the employed transfection protocol has a confounding impact on the performed experiments. Further, the standard deviations mentioned in the figure legend are not included in the actual figure.

Response: Although all fibroblasts depicted in Figure 2b and 3b (Figure 2c and Figure 3c in the revised version) came from the same sources (either P1 or HC cultures), the interventions that they were exposed to (e.g., nucleofection/electroporation for the transfected ones; none for the un-transfected), and more importantly the timing of the imaging/aim of the experiment, account for the image discrepancies described by the Reviewers. In Figure 2b (top panel, un-transfected/untreated P1 and HC fibroblast), images were collected 20min after cells were seeded as we focused our analysis on filopodia formation that was best imaged at this time point and resulted to be highly prevalent in P1 fibroblasts in comparison to the HC. In Figure 3b (nucleofection/electroporated cells), images were collected 120min after seeding as we focused our aim on lamellipodia formation that was best imaged at this time point and resulted to be highly present in the WT ARPC5 efficiently transfected cells, but virtually absent in the empty vector transfected cells. Representative low power field images collected 20min (as in Figure 2c) or 120min (as in Figure 3c) after seeding are included below (20min white arrows, filopodia; 120min white arrows, lamellipodia). Altogether, different timepoints (i.e., 20min vs. 120min) represent different stages of cell adhesion, spreading and actin polymerization and should be compared within, but not across, time points. To avoid data misinterpretation by the readers, the timing of the imaging is now further clarified in the material & methods section and the figure legends.

The mean +/- SD for each time point in Figures 2c and d, and 3d are now more clearly visible and further clarified in their figure legends.

The authors state on page 8:

“Treatment of HC’s fibroblasts with the ARP2/3 inhibitor CK-666 resulted in reduced spreading capacity mimicking P1’s fibroblasts behavior (Fig.S7).”

This is true in principle. Nevertheless, the spreading behavior of the untreated HC fibroblasts judged by the cell index is quite different to what is shown in Figure 2B. It should also be noted that when looking at the fibroblast phenotype observed for P1 (Figure 2B), it is dramatically different from the CK-666 treated fibroblasts of a HC (Figure S7A).

Response: Related to the query above, HC and P1 fibroblast images depicted in Figure 2b (now Figure 2c) were collected on untreated cells 20min after they were seeded as this was the best time point for filopodia formation imaging. In contrast, the images of HC untreated or CK-666 treated fibroblasts depicted in Figure S7 were collected 120min after cells were seeded as this was the best time point for lamellipodia formation imaging. Representative low power field images of HC untreated and CK-666 treated fibroblasts collected 30min or 120min after seeding are included below (120min white arrows, lamellipodia). As stated in the previous query, different timepoints (i.e., 20-30min vs. 120min) represent different stages of cell adhesion, spreading and actin polymerization and should be compared within, but not across, time points. To avoid data misinterpretation by the readers, the timing of the imaging is now further clarified in the material & methods section and the figure legends.

On page 9 the authors state:

“ARPC5 staining localized to the nucleus, to scattered dots throughout the cytoplasm, and to the edge of the plasma membrane on rescued cells, compatible with expected ARP2/3 complex subcellular localization (Fig.3b, Fig.S9).” ARPC5 isoform-specific subcellular localization has been described and this statement could therefore be backed up with citations.

Millard et al, 2003 (DOI: 10.1002/cm.10087)

Faessler et al., 2023 (DOI: 10.1126/sciadv.add6495)

Response: ARPC5 was used as a search term in the human proteome database originally cited (https://www.nextprot.org/entry/NX_O15511/localization) and the revised version of the manuscript has been updated to indicate that. Besides, the references by Millard et al., and Faessler et al., are now added to the revised version as they further support the statement raised.

The authors state on page 11:

“In contrast, ARPC1B deficiency does not seem to affect ARPC5 expression levels, suggesting that ARPC5 deficiency might have a broader impact on biology and disease.”

Could the authors explain their argument further? Is this because ARPC5 might also affect ARPC1A containing complexes? Because then ARPC1B might in return affect ARPC5L containing complexes.

Response: As shown in Figures 2a, S6a and b from the original and the revised versions, primary cells from P1 completely lack ARPC5 expression and were also markedly deficient on ARPC1A and ARPC1B expression. In fact, all these defects were corrected upon rescued expression of WT ARPC5, as shown in Figures 3a and S9A. In addition, as previously described by others (JCI Insight. 2021;6[23]) and corroborated by us in Figure S6C, ARPC1B deficiency does not directly impact on ARPC5 levels. Altogether, these data strongly suggests that in ARPC1B deficiency, ARPC1B is the only Arp2/3 protein decreased while in ARPC5 deficiency, ARPC5 plus two other Arp2/3 complex proteins, ARPC1A and ARPC1B, are markedly diminished. In addition, when Leung et al. (JCI Insight 2021) explored B cell constitutive signaling in ARPC1B deficient cells, they found markedly increased ARPC1A accumulation, likely supporting the rescue of particular, but not all, ARPC1B-dependent functions. Altogether, and as

suggested by the Reviewers, these facts allow us to conclude that ARPC5 deficiency has a broader impact on biology and disease than ARPC1B deficiency.

While we effectively proved that ARPC5 deficiency affect the quality (mobility shift migration) and quantity of ARPC5- and ARPC2-containing Arp2/3 complexes (Figure 3b), the impact on ARPC1B-containing Arp2/3 complexes seems more limited and likely confined to a lower quantity (Figure S9B). As we did not experimentally evaluated ARPC1A- and ARPC5L-containing complexes in ARPC5 deficiency, we can only speculate about its effect. In that context, we agree with the Reviewers that the simultaneous decrease on ARPC5, ARPC1A and ARPC1B might impact ARPC5L-containing complexes. However, as ARPC5 deficiency in human vs. murine or rat cells seems to have a different behavior on ARPC5L regulation (e.g., virtually no effect in human cells as shown in our work, doubling expression in murine and rat cells as shown in Faessler et al., 2023 (DOI: 10.1126/sciadv.add6495), it will be more difficult to predict the type or magnitude of such effect.

The statement on page 12: “Branched actin is the predominant form of actin organization in lamellipodia, podosomes and invadopodia, all required for cell motility²¹.” This is not completely true as cell motility is possible without branched actin networks, for example in B16-F1 cells devoid of WAVE1/WAVE2, which do not exhibit lamellipodia (Tang et al., 2020, DOI: 10.1091/mbc.E19-12-0705). The same is true for leukocytes lacking one of the WRC subunits Hem1 (Leithner, Eichner et al., 2016 <https://doi.org/10.1038/ncb3426>).

Response: We appreciate the Reviewers’ comment. The phrase was changed to “Branched actin is the predominant form of actin organization in lamellipodia, podosomes and invadopodia, all required, but not indispensable, for cell motility.”

The authors state:

“Whether Arp2/3 complex integrity in these scenarios is maintained by alternative isoforms of ARPC1 and ARPC5 or by non-canonical conformations of the complex lacking these subunits²⁶ is yet to be established.” We have already shown that complete Arp2/3 complexes containing specifically ARPC5L are devoid of ARPC5 (Faessler et al., 2023, DOI: 10.1126/sciadv.add6495).

Figure S9B: Why has ARPC1b been chosen for this blot? This does not represent all ARP2/3 complexes due to ARPC1 isoform diversity and is inconsistent with the main figures.

Response: As ARPC1B is markedly decreased in ARPC5 deficiency and the only other human primary genetic disease directly impacting the Arp2/3 complex genes/proteins, in our perspective it was a proper experiment to add. The ARPC1B-containing Arp2/3 complexes were also explored to complement and expand the ARPC5- and ARPC2-containing Arp2/3 complexes investigated and depicted in Figure 3b. Interestingly, 3 different patterns of Arp2/3 complex were observed when each of the antibodies were tested, further highlighting the diversity and plasticity of Arp2/3 complex protein interactions.

Minor comments

- Page 8: “Wild-type ARPC5 expression restored abnormal findings”: do the authors maybe mean “rescue abnormal findings”

Response: We appreciate the Reviewers’ comment. The section title was corrected as suggested.

- Figure 1D: the illustration of the branch junction does look off, with respect to how the different subunits are arranged (see for example recent work by the Pollard and Nolen labs on how the Arp2/3 complex is arranged within the branch junction). Maybe consider exchanging this panel.

Response: We originally included Figure 1D depicting a cartoon of the Arp2/3 complex and function as a background reference for the readers; as both Reviewer 1 and Reviewers 3 found it odd, we decided to remove it from the revised version.

- Figure 2A: the labeling of the native blot is confusing with “healthy control” being written side-by-side. Please arrange “healthy control” to be in one column.

Response: Healthy control lanes 1-3 were re-labeled vertically in figure 2a; healthy control labels of the native blots (figure 2b in the revised manuscript) are now arranged in one column.

- Figure 2B (right panels) and C: Panels lack scale bars.

Response: Scale bars were added to Figures 2d and e in the revised version.

- Figure 3C: Scale bars are missing

Response: For the re-labeled Figures 3e and 2e (previous 3C and 2C, both wound healing assays), the most relevant measure to consider is the scratch width made by a pipette tip already described in the Materials and Methods section. The information on the 20 μ l Thermo Scientific Art tips, tip width= 0.94 mm is now included in the revised version.

- Figure S8: The scale indication is hard to interpret. Please provide clearly visible scale bars. The corresponding supplementary materials and methods section is ambiguous about how the displayed specimen were prepared: “Poly-L-lysine-coated glass coverslips or glow-discharged carbon-coated glass coverslips were prepared, and cells were added to the coverslips and fixed again in 2% glutaraldehyde for 15 min.” Which of these preparations is shown.

Response: Scale bars were enhanced in Figure S8 to improve visibility; the Materials and Methods section associated to it was also clarified.

- There seems to be a typo in the following sentence: “The coverslip was mounted on an SEM stub and coated with a thin layer of approximately 20-30 nm of a gold-palladium in a vacuum evaporator for conductivity under electron beam and then imaged with Hitachi S-4500 field emission scanning electron microscope (FESEM).

Response: In the revised supplementary methods this sentence was revised and updated to: “The polyester membrane filters were mounted on SEM stubs and coated with a thin layer of approximately 20-30 nm of a gold-palladium in the Edwards Emitech K575X Peltier Cooled Vacuum Evaporator for conductivity under electron beam and then imaged with a Hitachi S-4500 field

emission scanning electron microscope (FESEM) and image software Quartz PCI (Quartz Imaging Corporation, version 9). “

- Figure 3A and S9B: the authors might want to consider merging these figures.

Response: For space reasons and to avoid further overcrowding, we kept the figures separate.

Methods and Supplementary Methods

- State the amount of protein loaded onto PAGE gels.

Response: The amount of protein loaded onto PAGE gels is now included in the Methods section.

- Where possible, provide software version numbers and citations.

Response: Information on software used for data analysis is provided in the Methods section of the main manuscript, in supplementary methods, and in Nature’s portfolio reporting summary.

- Please provide manufacturers for all products mentioned in the methods section.

Response: Manufacturer’s information is provided in the Methods section.

- Please state which milk powder and which buffer were combined to produce the blocking buffer, which buffer was used for washing and which buffer was used for diluting antibodies for Western blotting.

Response: The milk powder and buffer used to produce the blocking buffer, antibody dilution buffer, and washing buffer for Western blotting are now specified in the Methods section.

- Please state the composition of the blocking buffer employed for immunocytochemistry in the Methods section. Is it the same one as described in the Supplementary Methods?

Response: The composition of the blocking buffer (10% FBS and 0.1% Triton X-100 in PBS) is provided in the Methods section.

- Full (non-cropped) Western blots should be provided as a supplementary figure.

Response: Uncropped versions of blots presented in the main manuscript or as supplementary information are included in the Source Data File, in compliance with Nature Communications reporting requirements.

*Kind Regards,
Florian Fäßler and Florian Schur*

REVIEWERS' COMMENTS

Reviewer #1 (Remarks to the Author):

The authors have responded adequately to my concerns. No further comments.

Reviewer #2 (Remarks to the Author):

The authors have addressed all points.

Reviewer #3 (Remarks to the Author):

The authors have provided extensive replies to the reviewer's comments, which has substantially improved the manuscript.

We would ask the authors for one small addition.

The requested quantification of western blots has been provided, but as it appears for only one blot (which is referred to as representative for two independent experiments). For completeness, we ask to provide the blots for these second experiments as well in the source file.

As two independent experiments are not sufficient for proper statistical analysis of changes, providing the blot of the second experiment in the source file allows the readers at least to visually compare these blots.

REVIEWERS' COMMENTS

Reviewer #1 (Remarks to the Author):

The authors have responded adequately to my concerns. No further comments.

Response: We highly appreciate the Reviewer's comment

Reviewer #2 (Remarks to the Author):

The authors have addressed all points.

Response: We highly appreciate the Reviewer's comment

Reviewer #3 (Remarks to the Author):

The authors have provided extensive replies to the reviewer's comments, which has substantially improved the manuscript.

Response: We highly appreciate the Reviewers' comment

We would ask the authors for one small addition.

The requested quantification of western blots has been provided, but as it appears for only one blot (which is referred to as representative for two independent experiments). For completeness, we ask to provide the blots for these second experiments as well in the source file.

As two independent experiments are not sufficient for proper statistical analysis of changes, providing the blot of the second experiment in the source file allows the readers at least to visually compare these blots.

Response: The "2nd independent experiment" western blots not shown in the manuscript are now included in the Data Source file following Nature Communications policies.